# SimPer: Simple Self-Supervised Learning of Periodic Targets

**Yuzhe Yang[1], Xin Liu[2], Jiang Wu[3], Silviu Borac[3], Dina Katabi[1], Ming-Zher Poh[3], Daniel McDuff[2,3]**
[1]MIT CSAIL     [2]University of Washington     [3]Google

## Abstract

From human physiology to environmental evolution, important processes in nature often exhibit meaningful and strong *periodic* or *quasi-periodic* changes. Due to their inherent label scarcity, learning useful representations for periodic tasks with limited or no supervision is of great benefit. Yet, existing self-supervised learning (SSL) methods overlook the intrinsic periodicity in data, and fail to learn representations that capture periodic or frequency attributes. In this paper, we present `SimPer`, a simple contrastive SSL regime for learning periodic information in data. To exploit the periodic inductive bias, `SimPer` introduces customized augmentations, feature similarity measures, and a generalized contrastive loss for learning efficient and robust periodic representations. Extensive experiments on common real-world tasks in human behavior analysis, environmental sensing, and healthcare domains verify the superior performance of `SimPer` compared to state-of-the-art SSL methods, highlighting its intriguing properties including better data efficiency, robustness to spurious correlations, and generalization to distribution shifts. Code and data are available at: https://github.com/YyzHarry/SimPer.

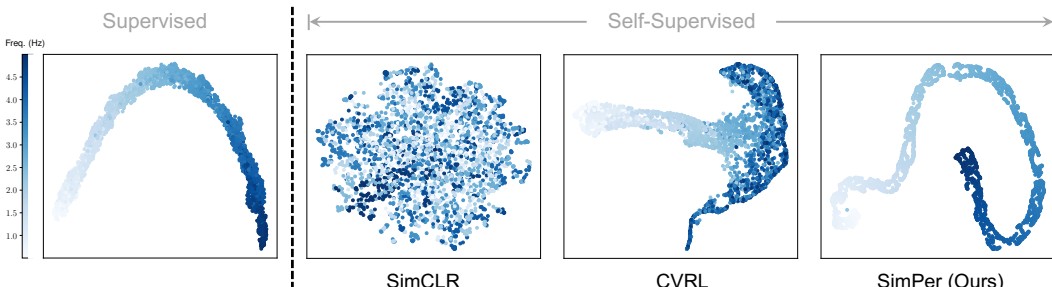

Figure 1: Learned representations of different methods on a periodic learning dataset, RotatingDigits (details in Section 4). Existing self-supervised learning schemes fail to capture the underlying periodic or frequency information in data. In contrast, `SimPer` learns robust periodic representations with high frequency resolution.

## 1 Introduction

Practical and important applications of machine learning in the real world, from monitoring the earth from space using satellite imagery (Espeholt et al., 2021) to detecting physiological vital signs in a human being (Luo et al., 2019), often involve recovering *periodic* changes. In the **health** domain, learning from video measurement has shown to extract (quasi-)periodic vital signs including atrial fibrillation (Yan et al., 2018), sleep apnea episodes (Amelard et al., 2018) and blood pressure (Luo et al., 2019). In the **environmental remote sensing** domain, periodic learning is often needed to enable nowcasting of environmental changes such as precipitation patterns or land surface temperature (Sønderby et al., 2020). In the **human behavior analysis** domain, recovering the frequency of changes or the underlying temporal morphology in human motions (e.g., gait or hand motions) is crucial for those rehabilitating from surgery (Gu et al., 2019), or for detecting the onset or progression of neurological conditions such as Parkinson's disease (Liu et al., 2022; Yang et al., 2022b).

While learning periodic targets is important, labeling such data is typically challenging and resource intensive. For example, if designing a method to measure heart rate, collecting videos with highly synchronized gold-standard signals from a medical sensor is time consuming, labor intensive, and requires storing privacy sensitive bio-metric data. Fortunately, given the large amount of unlabeled data, *self-supervised learning* that captures the underlying periodicity in data would be promising.

Yet, despite the great success of self-supervised learning (SSL) schemes on solving discrete classification or segmentation tasks, such as image classification (Chen et al., 2020; He et al., 2020), object detection (Xiao et al., 2021), action recognition (Qian et al., 2021), or semantic labeling (Hu et al., 2021), less attention has been paid to designing algorithms that capture periodic or quasi-periodic temporal dynamics from data. Interestingly, we highlight that existing SSL methods inevitably overlook the intrinsic periodicity in data: Fig. 1 shows the UMAP (McInnes et al., 2018) visualization of learned representations on RotatingDigits, a toy periodic learning dataset that aims to recover the underlying rotation frequency of different digits (details in Section 4). As the figure shows, state-of-the-art (SOTA) SSL schemes fail to capture the underlying periodic or frequency information in the data. Such observations persist across tasks and domains as we show later in Section 4.

To fill the gap, we present `SimPer`, a simple self-supervised regime for learning periodic information in data. Specifically, to leverage the temporal properties of periodic targets, `SimPer` first introduces a *temporal self-contrastive learning* framework, where positive and negative samples are obtained through *periodicity-invariant* and *periodicity-variant* augmentations from the **same** input instance. Further, we identify the problem of using conventional feature similarity measures (e.g., $\cos(\cdot)$) for periodic representation, and propose *periodic feature similarity* to explicitly define how to measure similarity in the context of periodic learning. Finally, to harness the intrinsic *continuity* of augmented samples in the frequency domain, we design a *generalized contrastive loss* that extends the classic InfoNCE loss to a soft regression variant that enables contrasting over continuous labels (frequency).

To support practical evaluation of SSL of periodic targets, we benchmark `SimPer` against SOTA SSL schemes on six diverse periodic learning datasets for common real-world tasks in human behavior analysis, environmental remote sensing, and healthcare. Rigorous experiments verify the robustness and efficiency of `SimPer` on learning periodic information in data. Our contributions are as follows:

- We identify the limitation of current SSL methods on periodic learning tasks, and uncover intrinsic properties of learning periodic dynamics with self-supervision over other mainstream tasks.
- We design `SimPer`, a simple & effective SSL framework that learns periodic information in data.
- We conduct extensive experiments on six diverse periodic learning datasets in different domains: human behavior analysis, environmental sensing, and healthcare. Rigorous evaluations verify the superior performance of `SimPer` against SOTA SSL schemes.
- Further analyses reveal intriguing properties of `SimPer` on its data efficiency, robustness to spurious correlations & reduced training data, and generalization to unseen targets.

## 2 RELATED WORK

**Periodic Tasks in Machine Learning.** Learning or recovering periodic signals from high dimensional data is prevailing in real-world applications. Examples of periodic learning include recovering and magnifying physiological signals (e.g., heart rate or breathing) (Wu et al., 2012), predicting weather and environmental changes (e.g., nowcasting of precipitation or land surface temperatures) (Sønderby et al., 2020; Espeholt et al., 2021), counting motions that are repetitive (e.g., exercises or therapies) (Dwibedi et al., 2020; Ali et al., 2020), and analyzing human behavior (e.g., gait) (Liu et al., 2022). To date, much prior work has focused on designing customized neural architectures (Liu et al., 2020; Dwibedi et al., 2020), loss functions (Starke et al., 2022), and leveraging relevant learning paradigms including transfer learning (Lu et al., 2018) and meta-learning (Liu et al., 2021) for periodic learning in a *supervised* manner, with high-quality labels available. In contrast to these past work, we aim to learn robust & efficient periodic representations in a *self-supervised* manner.

**Self-Supervised Learning.** Learning with self-supervision has recently attracted increasing interests, where early approaches mainly rely on pretext tasks, including exemplar classification (Dosovitskiy et al., 2014), solving jigsaw puzzles (Noroozi & Favaro, 2016), object counting (Noroozi et al., 2017), clustering (Caron et al., 2018), and predicting image rotations (Gidaris et al., 2018). More recently, a line of work based on contrastive losses (Oord et al., 2018; Tian et al., 2019; Chen et al., 2020; He et al., 2020) shows great success in self-supervised representations, where similar embeddings are learned for different views of the same training example (*positives*), and dissimilar embeddings for different training examples (*negatives*). Successful extensions have been made to temporal learning domains including video understanding (Jenni et al., 2020) or action classification (Qian et al., 2021). However, current SSL methods have limitations in learning periodic information, as the periodic inductive bias is often overlooked in method design. Our work extends existing SSL frameworks to periodic tasks, and introduces new techniques suitable for learning periodic targets.

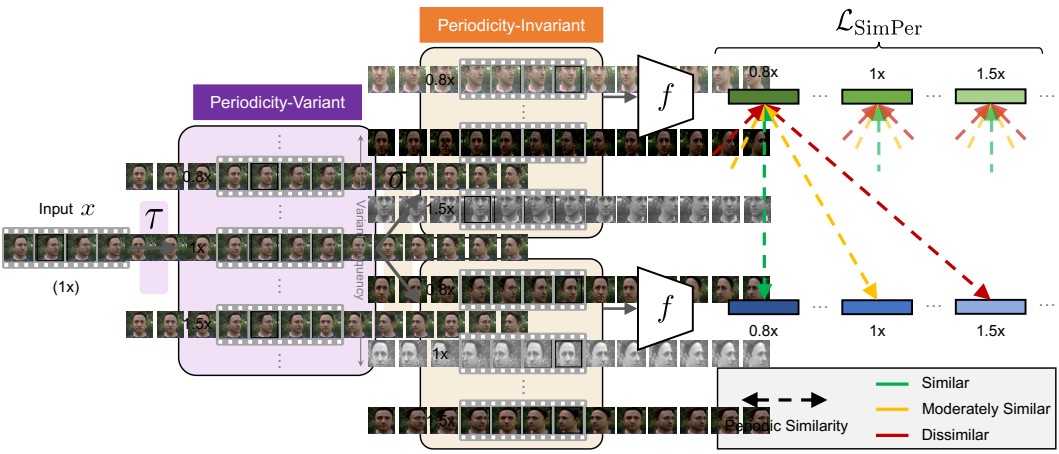

Figure 2: **An overview of the** `SimPer` **framework.** Input sequence is first passed through periodicity-variant transformations $\tau(\cdot)$ to create a series of speed (frequency) changed samples, where each augmented sample exhibits different underlying periodic signals due to the altered frequency, and can be treated as *negative* examples for each other. The augmented series are then passed through two sets of periodicity-invariant transformations $\sigma(\cdot)$ to create different invariant views (*positives*). All samples are then encoded in the feature space through a shared encoder $f(\cdot)$. The `SimPer` loss is calculated by contrasting over continuous speed (frequency) labels of different feature vectors, using customized periodic feature similarity measures.

## 3 THE SIMPER FRAMEWORK

When learning from periodic data in a self-supervised manner, a fundamental question arises:

*How do we design a self-supervised task such that **periodic** inductive biases are exploited?*

We note that periodic learning exhibits characteristics that are distinct from prevailing learning tasks. *First*, while most efforts on exploring invariances engineer transformations in the spatial (e.g., image recognition) or temporal (e.g., video classification) domains, dynamics in the **frequency** domain are essential in periodic tasks, which has implications for how we design (in)variances. *Second*, unlike conventional SSL where a cosine distance is typically used for measuring feature similarity, representations learned for repetitious targets inherently possess periodicity that is insensitive to certain shifts (e.g., shifts in feature index), which warrants new machinery for measuring *periodic* similarity. *Third*, labels of periodic data have a natural ordinality and continuity in the frequency domain, which inspires the need for strategies beyond instance discrimination, that contrast over *continuous* targets.

We present `SimPer` (Simple SSL of Periodic Targets), a unified SSL framework that addresses each of the above limitations. Specifically, `SimPer` first introduces a ***temporal self-contrastive learning*** scheme, where we design *periodicity-invariant* and *periodicity-variant* augmentations for the *same* input instance to create its effective *positive* and *negative* views in the context of periodic learning (Section 3.1). Next, `SimPer` presents ***periodic feature similarity*** to explicitly define how one should measure the feature similarity when the learned representations inherently possess periodic information (Section 3.2). Finally, in order to exploit the continuous nature of augmented samples in the frequency domain, we propose a ***generalized contrastive loss*** that extends the classic InfoNCE loss (Oord et al., 2018) from *discrete* instance discrimination to *continuous* contrast over frequencies, which takes into account the meaningful distance between continuous labels (Section 3.3).

### 3.1 TEMPORAL SELF-CONTRASTIVE LEARNING FRAMEWORK

**Problem Setup.** Let $\mathcal{D} = \{(\mathbf{x}_i)\}_{i=1}^{N}$ be the unlabeled training set, where $\mathbf{x}_i \in \mathbb{R}^D$ denotes the input sequence. We denote as $\mathbf{z} = f(\mathbf{x}; \theta)$ the representation of $\mathbf{x}$, where $f(\cdot; \theta)$ is parameterized by a deep neural network with parameter $\theta$. To preserve the full temporal dynamics and information, $\mathbf{z}$ typically extracts *frame-wise* feature of $\mathbf{x}$, i.e., $\mathbf{z}$ has the same length as input $\mathbf{x}$.

As motivated, ***frequency*** information is most essential when learning from periodic data. Precisely, augmentations that change the underlying frequency effectively alter the identity of the data (periodicity), and vice versa. This simple insight has implications for how we design proper (in)variances.

**Periodicity-Variant Augmentations.** We construct negative views of data through transformations in the frequency domain. Specifically, given input sequence $\mathbf{x}$, we define periodicity-variant aug-

mentations $\tau \in \mathcal{T}$, where $\mathcal{T}$ represents the set of transformations that change $\mathbf{x}$ with an *arbitrary* speed that is feasible under the Nyquist sampling theorem. As Fig. 2 shows, SimPer augments $\mathbf{x}$ by $M$ times, obtaining a series of *speed (frequency)* changed samples $\{\tau_1(\mathbf{x}), \tau_2(\mathbf{x})\}, \ldots, \tau_M(\mathbf{x})\}$, whose relative speeds satisfy $s_1 < s_2 < \ldots < s_M, s_i \propto \text{freq}(\tau_i(\mathbf{x}))$. Such augmentation effectively changes the underlying periodic targets with shifted frequencies, thus creating different *negative* views. Therefore, although the original target frequency is unknown, we effectively devise *pseudo speed (frequency) labels* for unlabeled $\mathbf{x}$. In practice, we limit the speed change range to be within $[s_{\min}, s_{\max}]$, ensuring the augmented sequence is longer than a fixed length in the time dimension.

**Periodicity-Invariant Augmentations.** We further define periodicity-invariant augmentation $\sigma \in \mathcal{S}$, where $\mathcal{S}$ denotes the set of transformations that do not change the *identity* of the original input. When the set is finite, i.e., $\mathcal{S} = \{\sigma_1, \ldots, \sigma_k\}$, we have $\text{freq}(\sigma_i(\mathbf{x})) = \text{freq}(\sigma_j(\mathbf{x})), \forall i, j \in [k]$. Such augmentations can be used to learn invariances in the data from the perspective of periodicity, creating different *positive* views. Practically, we leverage spatial (e.g., crop & resize) and temporal (e.g., reverse, delay) augmentations to create different views of the same instance (see Fig. 2).

**Temporal Self-Contrastive Learning.** Unlike conventional contrastive SSL schemes where augmentations are exploited to produce invariances, i.e., creating different positive views of the data, SimPer introduces periodicity-variant augmentations to explicitly model what *variances* should be in periodic learning. Concretely, negative views are no longer from other *different* instances, but directly from the *same* instance itself, realizing a ***self-contrastive*** scheme. Table 1 details the differences.

Table 1: Differences of view constructions.

| Algorithm | Positives | Negatives |
|---|---|---|
| Conventional SSL methods | **Instance:** Same **Aug.:** Invariant | **Instance:** Different **Aug.:** Invariant |
| SimPer | **Instance:** Same **Aug.:** Period.-Invariant | **Instance:** Same **Aug.:** Period.-Variant |

We highlight the benefits of using the self-contrastive framework. First, it provides *arbitrarily large* negative sample sizes, as long as the Nyquist sampling theorem is satisfied. This makes SimPer not dependent on the actual training set size, and enables effective contrasting even under limited data scenarios. We show in Section 4.2 that when drastically reducing the dataset size to only $5\%$ of the total samples, SimPer still works equally well, substantially outperforming supervised counterparts. Second, our method naturally leads to *hard* negative samples, as periodic information is directly being contrasted, while unrelated information (e.g., frame appearance) are maximally preserved across negative samples. This makes SimPer robust to spurious correlations in data (Section 4.5).

## 3.2 FEATURE SIMILARITY IN THE CONTEXT OF PERIODIC LEARNING

We identify that *feature similarity measures* are also different in the context of periodic representations. Consider sampling two short clips $\mathbf{x}_1, \mathbf{x}_2$ from the same input sequence, but with a frame shift $t$. Assume the frequency does not change within the sequence, and its period $T > t$. Since the underlying information does not change, by definition their features should be close in the embedding space (i.e., high feature similarity). However, due to the shift in time, when extracting frame-level feature vectors, the indexes of the feature representations (which represent different time stamps) will no longer be aligned. In this case, if directly using a cosine similarity as defined in conventional SSL literature, the similarity score would be low, despite the fact that the actual similarity is high.

**Periodic Feature Similarity.** To overcome this limitation, we propose to use *periodic* feature similarity measures in SimPer. Fig. 3 highlights the properties and differences between conventional feature similarity measures and the desired similarity measure in periodic learning. Specifically, existing SSL methods adopt similarity measures that emphasize strict "closeness" between two feature vectors, and are sensitive to shifted or reversed feature indexes. In contrast, when aiming for learning periodic features, a proper periodic feature measure should retain high similarity for features with shifted (sometimes reversed) indexes, while also capturing a continuous similarity change when the feature frequency varies, due to the meaningful distance in the frequency domain.

**Concrete Instantiations.** We provide two practical instantiations to effectively capture the periodic feature similarity. Note that these instantiations can be easily extended to high-dimensional features (in addition to the time dimension) by averaging across other dimensions.

- *Maximum cross-correlation (**MXCorr**)* measures the maximum similarity as a function of offsets between signals (Welch, 1974), which can be efficiently computed in the frequency domain.

- *Normalized power spectrum density (**nPSD**)* calculates the distance between the normalized PSD of two feature vectors. The distance can be a cosine or $L_2$ distance (details in Appendix D.4.3).

Figure 3: Differences between **(a)** conventional feature similarity, and **(b)** *periodic* feature similarity. A proper periodic feature similarity measure should induce high similarity for features with shifted (sometimes reversed) indexes, while capturing a continuous similarity change when the feature frequency varies.

### 3.3 GENERALIZED CONTRASTIVE LOSS WITH CONTINUOUS TARGETS

Motivated by the fact that the augmented views are ***continuous*** in frequency, where the *pseudo speed labels* $\{s_i\}_{i=1}^{M}$ are known through augmentation (i.e., a view at $1.1\times$ is more similar to the original than that at $2\times$), we relax and extend the original InfoNCE contrastive loss (Oord et al., 2018) to a soft variant, where it generalizes from discrete instance discrimination to continuous targets.

**From Discrete Instance Discrimination to Continuous Contrast.** The classic formulation of the InfoNCE contrastive loss for each input sample $\mathbf{x}$ is written as

$$\mathcal{L}_{\text{InfoNCE}} = -\log \frac{\exp(\text{sim}(\mathbf{z}, \widehat{\mathbf{z}})/\nu)}{\sum_{\mathbf{z}' \in \mathcal{Z} \setminus \{\mathbf{z}\}} \exp(\text{sim}(\mathbf{z}, \mathbf{z}')/\nu)}, \tag{1}$$

where $\widehat{\mathbf{z}} = f(\widehat{\mathbf{x}})$ ($\widehat{\mathbf{x}}$ is the positive pair of $\mathbf{x}$ obtained through augmentations), $\mathcal{Z}$ is the set of features in current batch, $\nu$ is the temperature constant, and $\text{sim}(\cdot)$ is usually instantiated by a dot product. Such format indicates a *hard* classification task, where target label is $1$ for positive pair and $0$ for all negative pairs. However, negative pairs in `SimPer` inherently possess a meaningful distance, which is reflected by the similarity of their relative speed (frequency). To capture this intrinsic continuity, we consider the contributions from *all* pairs, with each scaled by the *similarity* in their labels.

**Generalized InfoNCE Loss.** For an input sample $\mathbf{x}$, `SimPer` creates $M$ variant views with different speed labels $\{s_i\}_{i=1}^{M}$. Given the features of two sets of invariant views $\{\mathbf{z}_i\}_{i=1}^{M}, \{\mathbf{z}'_i\}_{i=1}^{M}$, we have

$$\mathcal{L}_{\text{SimPer}} = \sum_i \ell_{\text{SimPer}}^i = \sum_i - \sum_{j=1}^{M} \frac{\exp(w_{i,j})}{\sum_{k=1}^{M} \exp(w_{i,k})} \log \frac{\exp(\text{sim}(\mathbf{z}_i, \mathbf{z}'_j)/\nu)}{\sum_{k=1}^{M} \exp(\text{sim}(\mathbf{z}_i, \mathbf{z}'_k)/\nu)}, \quad w_{i,j} := \text{sim}_{\text{label}}(s_i, s_j),$$

$$\tag{2}$$

where $\text{sim}(\cdot)$ denotes the *periodic* feature similarity as described previously, and $\text{sim}_{\text{label}}(\cdot)$ denotes the *continuous* label similarity measure. In practice, $\text{sim}_{\text{label}}(\cdot)$ can be simply instantiated as inverse of the $L_1$ or $L_2$ label difference (e.g., $1/|s_i - s_j|$).

**Interpretation.** $\mathcal{L}_{\text{SimPer}}$ is a simple generalization of the InfoNCE loss from discrete instance discrimination (single target classification) to a weighted loss over all augmented pairs (soft regression variant), where the soft target $\exp(w_{i,j})/\sum_k \exp(w_{i,k})$ is driven by the *label* (speed) similarity $w_{i,j}$ of each pair. Note that when the label becomes discrete (i.e., $w_{i,j} \in \{0, 1\}$), $\mathcal{L}_{\text{SimPer}}$ degenerates to the original InfoNCE loss. We demonstrate in Appendix D.4.4 that such continuity modeling via a generalized loss helps achieve better downstream performance than simply applying InfoNCE.

## 4 EXPERIMENTS

**Datasets.** We perform extensive experiments on six datasets that span different domains and tasks. Complete descriptions of each dataset are in Appendix B, Fig. 7, and Table 11.

- ***RotatingDigits** (Synthetic Dataset)* is a toy periodic learning dataset consists of rotating MNIST digits (Deng, 2012). The task is to predict the underlying digit rotation frequency.
- ***SCAMPS** (Human Physiology)* (McDuff et al., 2022) consists of 2,800 synthetic videos of avatars with realistic peripheral blood flow. The task is to predict averaged heart rate from input videos.
- ***UBFC** (Human Physiology)* (Bobbia et al., 2019) contains 42 videos with synchronized gold-standard contact PPG recordings. The task is to predict averaged heart rate from input video clips.

Table 2: Feature evaluation results on RotatingDigits.

| | FFT | | 1-NN | |
|---|---|---|---|---|
| Metrics | MAE$^\downarrow$ | MAPE$^\downarrow$ | MAE$^\downarrow$ | MAPE$^\downarrow$ |
| SIMCLR (Chen et al., 2020) | 2.96 | 109.27 | 0.98 | 48.30 |
| MoCo v2 (He et al., 2020) | 2.83 | 90.78 | 0.62 | 32.74 |
| BYOL (Grill et al., 2020) | 2.20 | 78.43 | 0.46 | 22.08 |
| CVRL (Qian et al., 2021) | 1.69 | 49.09 | 0.38 | 14.41 |
| SIMPER | **0.22** | **16.49** | **0.09** | **4.51** |
| GAINS | **+1.47** | **+32.60** | **+0.29** | **+9.90** |

Table 3: Feature evaluation results on SCAMPS.

| | FFT | | 1-NN | |
|---|---|---|---|---|
| Metrics | MAE$^\downarrow$ | MAPE$^\downarrow$ | MAE$^\downarrow$ | MAPE$^\downarrow$ |
| SIMCLR (Chen et al., 2020) | 27.48 | 38.39 | 34.09 | 40.79 |
| MoCo v2 (He et al., 2020) | 28.16 | 40.23 | 35.61 | 42.47 |
| BYOL (Grill et al., 2020) | 26.15 | 37.34 | 32.77 | 38.26 |
| CVRL (Qian et al., 2021) | 27.67 | 38.80 | 33.32 | 39.54 |
| SIMPER | **14.45** | **22.09** | **13.75** | **18.64** |
| GAINS | **+11.70** | **+15.25** | **+19.02** | **+19.62** |

Table 4: Feature evaluation results on UBFC.

| | FFT | | 1-NN | |
|---|---|---|---|---|
| Metrics | MAE$^\downarrow$ | MAPE$^\downarrow$ | MAE$^\downarrow$ | MAPE$^\downarrow$ |
| SIMCLR (Chen et al., 2020) | 16.92 | 14.73 | 16.23 | 18.62 |
| MoCo v2 (He et al., 2020) | 14.64 | 13.17 | 15.12 | 16.56 |
| BYOL (Grill et al., 2020) | 17.86 | 16.90 | 18.13 | 19.34 |
| CVRL (Qian et al., 2021) | 11.75 | 10.67 | 12.36 | 13.38 |
| SIMPER | **8.78** | **7.46** | **8.92** | **10.21** |
| GAINS | **+2.97** | **+3.21** | **+3.44** | **+3.17** |

Table 5: Feature evaluation results on PURE.

| | FFT | | 1-NN | |
|---|---|---|---|---|
| Metrics | MAE$^\downarrow$ | MAPE$^\downarrow$ | MAE$^\downarrow$ | MAPE$^\downarrow$ |
| SIMCLR (Chen et al., 2020) | 23.70 | 22.07 | 29.48 | 31.44 |
| MoCo v2 (He et al., 2020) | 24.23 | 24.08 | 30.82 | 33.95 |
| BYOL (Grill et al., 2020) | 23.24 | 21.78 | 29.27 | 31.03 |
| CVRL (Qian et al., 2021) | 19.27 | 18.94 | 22.08 | 23.75 |
| SIMPER | **13.97** | **12.88** | **14.03** | **15.35** |
| GAINS | **+5.30** | **+6.06** | **+8.05** | **+8.40** |

Table 6: Feature evaluation results on Countix.

| | FFT | | 1-NN | |
|---|---|---|---|---|
| Metrics | MAE$^\downarrow$ | GM$^\downarrow$ | MAE$^\downarrow$ | GM$^\downarrow$ |
| SIMCLR (Chen et al., 2020) | 3.90 | 2.26 | 4.43 | 3.19 |
| MoCo v2 (He et al., 2020) | 3.75 | 2.18 | 3.96 | 3.04 |
| BYOL (Grill et al., 2020) | 3.26 | 1.87 | 3.72 | 2.66 |
| CVRL (Qian et al., 2021) | 2.81 | 1.38 | 3.15 | 2.12 |
| SIMPER | **2.06** | **0.98** | **2.76** | **1.84** |
| GAINS | **+0.75** | **+0.40** | **+0.99** | **+0.28** |

Table 7: Feature evaluation results on LST.

| | Linear Probing | | |
|---|---|---|---|
| Metrics | MAE$^\downarrow$ | MAPE$^\downarrow$ | $\rho^\uparrow$ |
| SIMCLR (Chen et al., 2020) | 5.12 | 0.20 | 0.89 |
| MoCo v2 (He et al., 2020) | 5.16 | 0.20 | 0.89 |
| BYOL (Grill et al., 2020) | 5.71 | 0.24 | 0.86 |
| CVRL (Qian et al., 2021) | 4.88 | **0.18** | **0.91** |
| SIMPER | **4.84** | **0.18** | 0.90 |
| GAINS | **+0.04** | **+0.00** | **−0.01** |

- **PURE (Human Physiology)** (Stricker et al., 2014) contains 60 videos with synchronized gold-standard contact PPG recordings. The task is to predict averaged heart rate from input video clips.
- **Countix (Action Counting)**. The Countix dataset (Dwibedi et al., 2020) is a subset of the Kinetics (Kay et al., 2017) dataset annotated with segments of repeated actions and corresponding counts. The task is to predict the count number given an input video.
- **Land Surface Temperature (LST) (Satellite Sensing)**. LST contains hourly land surface temperature maps over the continental United States for 100 days (April 7th to July 16th, 2022). The task is to predict future temperatures based on past satellite measurements.

**Network Architectures.** We choose a set of logical architectures from prior work for our experiments. On RotatingDigits and SCAMPS, we employ a simple 3D variant of the CNN architecture as in (Yang et al., 2022a). Following (Liu et al., 2020), we adopt a variant of TS-CAN model for experiments on UBFC and PURE. Finally, on Countix and LST, we employ ResNet-3D-18 (He et al., 2016; Tran et al., 2018) as our backbone network. Implementations details are in Appendix C.

**Baselines.** We compare SimPer to SOTA SSL methods, including SimCLR (Chen et al., 2020), MoCo v2 (He et al., 2020), BYOL (Grill et al., 2020), and CVRL (Qian et al., 2021), as well as a supervised learning counterpart. We provide detailed descriptions in Appendix C.1.

**Evaluation Metrics.** To assess the prediction of continuous targets (e.g., frequency, counts), we use common metrics for regression, such as the mean-average-error (MAE), mean-average-percentage-error (MAPE), Pearson correlation ($\rho$), and error Geometric Mean (GM) (Yang et al., 2021).

## 4.1 MAIN RESULTS

We report the main results in this section for all datasets. Complete training details, hyper-parameter settings, and additional results are provided in Appendix C and D.

Table 8: Fine-tune evaluation results on all datasets. We first pre-train the feature encoder using different SSL methods, then fine-tune the whole network initialized with the pre-trained weights.

| Metrics | RotatingDigits MAE$^\downarrow$ | RotatingDigits MAPE$^\downarrow$ | SCAMPS MAE$^\downarrow$ | SCAMPS MAPE$^\downarrow$ | UBFC MAE$^\downarrow$ | UBFC MAPE$^\downarrow$ | PURE MAE$^\downarrow$ | PURE MAPE$^\downarrow$ | Countix MAE$^\downarrow$ | Countix GM$^\downarrow$ | LST MAE$^\downarrow$ | LST $\rho^\uparrow$ |
|---|---|---|---|---|---|---|---|---|---|---|---|---|
| SUPERVISED | 0.72 | 28.96 | 3.61 | 5.33 | 5.13 | 4.72 | 4.25 | 4.93 | 1.50 | 0.73 | 1.54 | **0.96** |
| SIMCLR (Chen et al., 2020) | 0.69 | 26.54 | 4.96 | 6.92 | 5.32 | 4.96 | 4.86 | 5.32 | 1.58 | 0.80 | 1.54 | 0.95 |
| MoCo v2 (He et al., 2020) | 0.64 | 24.73 | 5.33 | 7.24 | 5.05 | 4.64 | 4.97 | 5.60 | 1.54 | 0.79 | 1.53 | 0.95 |
| BYOL (Grill et al., 2020) | 0.39 | 20.91 | 3.49 | 5.27 | 5.51 | 5.07 | 4.28 | 4.97 | 1.47 | 0.71 | 1.62 | 0.92 |
| CVRL (Qian et al., 2021) | 0.34 | 18.82 | 5.52 | 7.34 | 5.07 | 4.70 | 4.19 | 4.71 | 1.48 | 0.71 | 1.49 | **0.96** |
| **SIMPER** | **0.20** | **14.33** | **3.27** | **4.89** | **4.24** | **3.97** | **3.89** | **4.01** | **1.33** | **0.59** | **1.47** | **0.96** |
| GAINS VS. SUPERVISED | +0.52 | +14.63 | +0.34 | +0.44 | +0.89 | +0.75 | +0.36 | +0.92 | +0.17 | +0.14 | +0.07 | +0.00 |

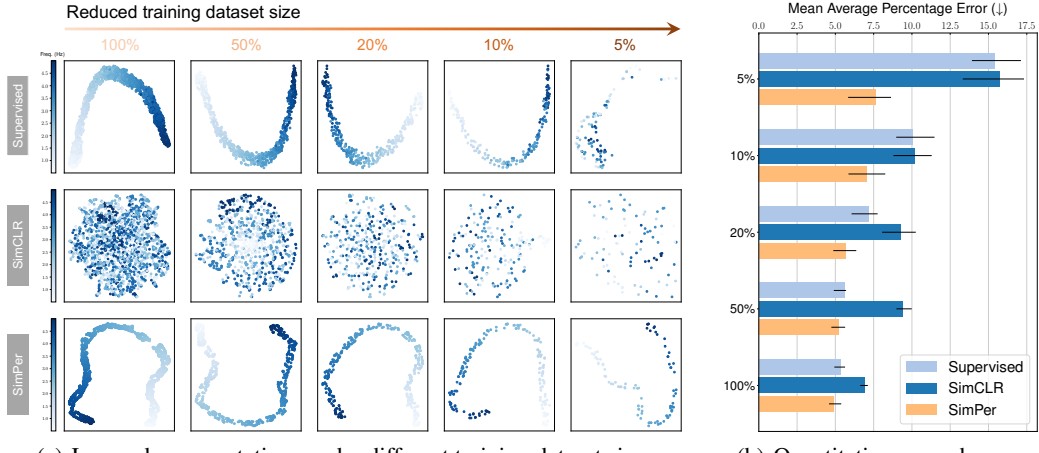

(a) Learned representations under different training dataset sizes

(b) Quantitative error changes

Figure 4: **Data efficiency analysis of** `SimPer`. **(a)** Learned representations of different algorithms on RotatingDigits when training dataset size reduces from $100\%$ to $5\%$. **(b)** The quantitative MAPE errors on SCAMPS with varying training dataset sizes. Complete quantitative results are provided in Appendix D.1.

**Feature Evaluation.** Following the literature (He et al., 2020; Chen et al., 2020), we first evaluate the representations learned by different methods. For dense prediction task (e.g., LST), we use the *linear probing* protocol by training a linear regressor on top of the fixed features. For tasks whose targets are frequency information, we directly evaluate the learned features using a Fourier transform (**FFT**) and a nearest neighbor classifier (**1-NN**). Table 2, 3, 4, 5, 6, 7 show the feature evaluation results of `SimPer` compared to SOTA SSL methods. As the tables confirm, across different datasets with various common tasks, `SimPer` is able to learn better representations that achieve the best performance. Furthermore, in certain datasets, the relative improvements are even larger than $50\%$.

**Fine-tuning.** Practically, to harness the power of pre-trained representations, fine-tuning the whole network with the encoder initialized using pre-trained weights is a widely adopted approach (He et al., 2020). To evaluate whether `SimPer` pre-training is helpful for each downstream task, we fine-tune the whole network and compare the final performance. The details of the setup for each dataset and algorithm can be found in Appendix C. As Table 8 confirms, across different datasets, `SimPer` consistently outperforms all other SOTA SSL methods, and obtains better results compared to the supervised baseline. This demonstrates that `SimPer` is able to capture meaningful periodic information that is beneficial to the downstream tasks.

## 4.2 DATA EFFICIENCY

In real-world periodic learning applications, data is often prohibitively expensive to obtain. To study the data efficiency of `SimPer`, we manually reduce the overall size of RotatingDigits, and plot the representations learned as well as the final fine-tuning accuracy of different methods in Fig. 4.

As the figure confirms, when the dataset size is large (e.g., using $100\%$ of the data), both supervised learning baseline and `SimPer` can learn good representations (Fig. 4(a)) and achieve low test errors

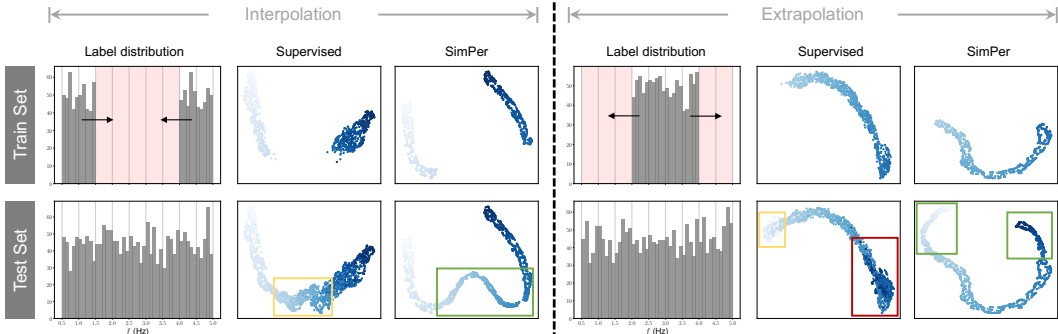

Figure 5: **Zero-shot generalization analysis.** We create training sets with missing target frequencies and keep test sets evenly distributed across the target range. Green regions indicate successful generalization with high frequency resolution. Yellow regions indicate successful generalization but with low frequency resolution. Red regions represent failed generalization. `SimPer` learns robust representations that generalize to unseen targets.

(Fig. 4(b)). However, when the training dataset size becomes smaller, the learned representations using supervised learning get worse, and eventually lose the frequency information and resolution when only 5% of the data is available. Correspondingly, the final error in this extreme case also becomes much higher. In contrast, even with small number of training data, `SimPer` can consistently learn the periodic information and maintain high frequency resolution, with significant performance gains especially when the available data amount is small.

## 4.3 TRANSFER LEARNING

We evaluate whether the self-supervised representations are transferable across datasets. We use UBFC and PURE, which share the same prediction task. Following (Chen et al., 2020), we fine-tune the pre-trained model on the new dataset, and compare the performance across both SSL and supervised methods. Table 9 reports the results, where in both cases, `SimPer` is able to achieve better final performance compared to supervised and SSL baselines, showing its ability to learn transferable periodic representations across different datasets.

Table 9: Transfer learning results.

| Metrics | UBFC → PURE | | PURE → UBFC | |
|---|---|---|---|---|
| | MAE↓ | MAPE↓ | MAE↓ | MAPE↓ |
| SUPERVISED | 7.83 | 8.85 | 3.15 | 3.11 |
| SIMCLR | 7.86 | 8.79 | 3.46 | 3.80 |
| **SIMPER** | **6.46** | **6.98** | **2.76** | **2.38** |
| GAINS | **+1.37** | **+1.87** | **+0.39** | **+0.73** |

## 4.4 ZERO-SHOT GENERALIZATION TO UNSEEN TARGETS

Given the continuous nature of the frequency domain, periodic learning tasks can (and almost certainly will) have unseen frequency targets during training, which motivates the need for target (frequency) extrapolation and interpolation. To investigate **zero-shot** generalization to unseen targets, we manually create training sets that have certain missing targets (Fig. 5), while making the test sets evenly distributed across the target range. As Fig. 5 confirms, in the interpolation case, both supervised learning and `SimPer` can successfully interpolate the missing targets. However, the quality of interpolation varies: For supervised learning, the frequency resolution is low within the interpolation range, resulting in mixed representations for a wide missing range. In contrast, `SimPer` learns better representations with higher frequency resolution, which has desirable discriminative properties.

Furthermore, in the extrapolation case, in the lower frequency range, both methods extrapolate reasonably well, with `SimPer` capturing a higher frequency resolution. However, when extrapolating to a higher frequency range, the supervised baseline completely fails to generalize, with learned features largely overlapping with the existing frequency targets in the training set. In contrast, `SimPer` is able to generalize robustly even for the higher unseen frequency range, demonstrating its effectiveness of generalization to distribution shifts and unseen targets. Quantitative results in Table 10 confirm the observations.

Table 10: Mean absolute error (MAE, ↓) results for zero-shot generalization analysis.

| | Interpolation | | Extrapolation | |
|---|---|---|---|---|
| | Seen | Unseen | Seen | Unseen |
| SUPERVISED | 0.09 | 0.85 | 0.03 | 1.74 |
| **SIMPER** | **0.05** | **0.07** | **0.02** | **0.02** |
| GAINS | **+0.04** | **+0.78** | **+0.01** | **+1.72** |

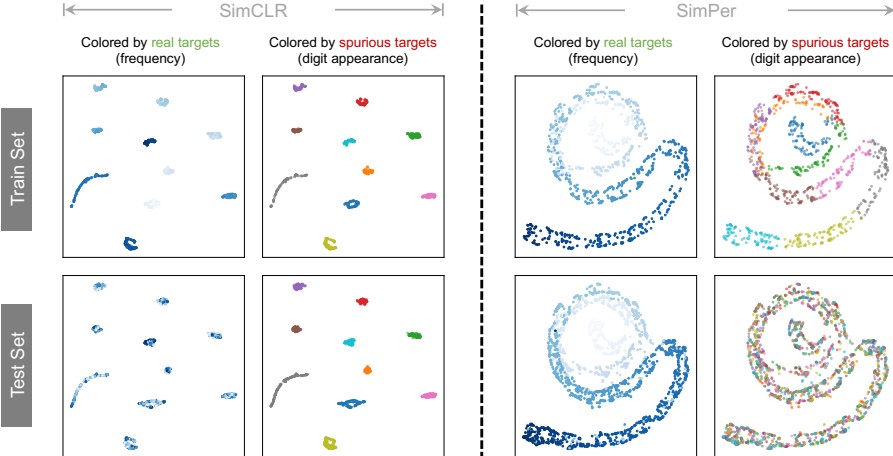

Figure 6: **Robustness to spurious correlations.** We make the target frequency spuriously correlated with digit appearances in training set, while removing this correspondence in test set. `SimPer` is able to capture underlying periodic information & learn robust representations that generalize. Quantitative results are in Appendix D.3.

## 4.5 ROBUSTNESS TO SPURIOUS CORRELATIONS

We show that `SimPer` is able to deal with spurious correlations that arise in data, while existing SSL methods often fail to learn generalizable features. Specifically, RotatingDigits dataset naturally has a spurious target: the digit appearance (number). We further enforce this information by coloring different digits with different colors as in (Arjovsky et al., 2019). We then construct a spuriously correlated training set by assigning a unique rotating frequency range to a specific digit, i.e., [0.5Hz, 1Hz] for digit 0, [1Hz, 1.5Hz] for digit 1, etc, while removing the spurious correlations in test set.

As Fig. 6 verifies, SimCLR is easy to learn information that is spuriously correlated in the training data, but not the actual target of interest (frequency). As a result, the learned representations do not generalize. In contrast, `SimPer` learns the underlying frequency information even in the presence of strong spurious correlations, demonstrating its ability to learn robust representations that generalize.

## 4.6 FURTHER ANALYSIS AND ABLATION STUDIES

**Amount of labeled data for fine-tuning (Appendix D.2).** We show that when the amount of labeled data is limited for fine-tuning, `SimPer` still substantially outperforms baselines by a large margin, achieving a $67\%$ relative improvement in MAE even when the labeled data fraction is only $5\%$.

**Ablation: Frequency augmentation range (Appendix D.4.1).** We study the effects of different speed (frequency) augmentation ranges when creating periodicity-variant views (Table 15). While a proper range can lead to certain gains, `SimPer` is reasonably robust to different choices.

**Ablation: Number of augmented views (Appendix D.4.2).** We investigate the influence of different number of augmented views (i.e., $M$) in `SimPer`. Interestingly, we find `SimPer` is surprisingly robust to different $M$ in a given range (Table 16), where larger $M$ often delivers better results.

**Ablation: Choices of different similarity metrics (Appendix D.4.3).** We explore the effects of different periodic similarity measures in `SimPer`, where we show that `SimPer` is robust to all aforementioned periodic similarity measures, achieving similar performances (Table 17).

**Ablation: Effectiveness of generalized contrastive loss (Appendix D.4.4).** We confirm the effectiveness of the generalized contrastive loss by showing its consistent performance gains across all six datasets, as compared to the vanilla InfoNCE loss (Table 18).

## 5 CONCLUSION

We present `SimPer`, a simple and effective SSL framework for learning periodic information from data. `SimPer` develops customized periodicity-variant and invariant augmentations, periodic feature similarity, and a generalized contrastive loss to exploit periodic inductive biases. Extensive experiments on different datasets over various real-world applications verify the superior performance of `SimPer`, highlighting its intriguing properties such as better efficiency, robustness & generalization.

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

## A    PSEUDO CODE FOR SIMPER

We provide the pseudo code of `SimPer` in Algorithm 1.

---

**Algorithm 1** Simple Self-Supervised Learning of Periodic Targets (`SimPer`)

---

**Input:** Unlabeled training set $\mathcal{D} = \{(\mathbf{x}_i)\}_{i=1}^{N}$, total training epochs $E$, periodicity-variant augmentations $\tau(\cdot)$, periodicity-invariant augmentations $\sigma(\cdot)$, number of variant views $M$, encoder $f$

**for** $e = 0$ **to** $E$ **do**

   **repeat**

      Sample a mini-batch $\{(\mathbf{x}^{(l)})\}_{l=1}^{n}$ from $\mathcal{D}$

      **for** $l = 1$ **to** $n$ (in parallel) **do**

        $\{\mathbf{x}_i^{(l)}\}_{i=1}^{M} \leftarrow \tau\left(\mathbf{x}^{(l)}\right)$              `// M variant views for `$l$`-th sample`

        $\{\mathbf{x}_i^{(l)}\}_{i=1}^{M},\ \{\mathbf{x}_i'^{(l)}\}_{i=1}^{M} \leftarrow \sigma\left(\{\mathbf{x}_i^{(l)}\}_{i=1}^{M}\right)$     `// two sets of invariant views`

        $\{\mathbf{z}_i^{(l)}\}_{i=1}^{M},\ \{\mathbf{z}_i'^{(l)}\}_{i=1}^{M} \leftarrow f\left(\{\mathbf{x}_i^{(l)}\}_{i=1}^{M}\right),\ f\left(\{\mathbf{x}_i'^{(l)}\}_{i=1}^{M}\right)$

        Calculate $\ell_{\text{SimPer}}^{(l)}$ for $l$-th sample using $\{\mathbf{z}_i^{(l)}\}_{i=1}^{M},\ \{\mathbf{z}_i'^{(l)}\}_{i=1}^{M}$ based on Eqn. (2)

      **end for**

      Calculate $\mathcal{L}_{\text{SimPer}}$ using $\frac{1}{n}\sum_{l=1}^{n}\ell_{\text{SimPer}}^{(l)}$ and do one training step

   **until** iterate over all training samples at current epoch $e$

**end for**

---

## B    DATASET DETAILS

In this section, we provide the detailed information of the six datasets we used in our experiments. Fig. 7 shows examples of each dataset, and Table 11 provides the statistics of each dataset.

**RotatingDigits** *(Synthetic Dataset).* We create RotatingDigits, a synthetic periodic learning dataset of rotating MNIST digits (Deng, 2012), where samples are created with the original digits rotating on a plain background at rotational frequencies between 0.5Hz and 5Hz. The training set consists of $1,000$ rotating video clips (100 samples per digit number), each sample with a frame length of 150 and a sampling rate of 30Hz. The test set consists of $2,000$ rotating video clips (200 samples per digit number).

**SCAMPS** *(Human Physiology).* The SCAMPS dataset (McDuff et al., 2022) contains $2,800$ synthetic videos of avatars with realistic peripheral blood flow and breathing. The faces are synthesized using a blendshape-based rig with $7,667$ vertices and $7,414$ polygons and the identity basis is learned from a set of high-quality facial scans. These texture maps were sampled from 511 facial scans of subjects. The distribution of gender, age and ethnicity of the subjects who provided the facial scans can be found in (Wood et al., 2021). Blood flow is simulated by adjusting properties of the physically-based shading material[1]. We randomly divide the whole dataset into training ($2,000$ samples), validation (400 samples), and test (400 samples) set. Each video clip has a frame length of 600 and a sampling rate of 30Hz.

**UBFC** *(Human Physiology).* The UBFC dataset (Bobbia et al., 2019) contains a total of $42$ videos from $42$ subjects. The videos were recorded using a `Logitech C920 HD Pro` at 30Hz. A pulse oximeter was used to obtain the gold-standard PPG data (30Hz). The raw resolution is $640 \times 480$ and videos are recorded in a uncompressed 8-bit RGB format. We postprocess the videos by cropping the face region and resizing them to $36 \times 36$. We manually divide each video into non-overlapping chunks (Liu et al., 2020) with a window size 180 frames (6 seconds). The resulting number of training and test samples are 518 and 106, respectively.

**PURE** *(Human Physiology).* The PURE dataset (Stricker et al., 2014) includes 60 videos from 10 subjects (8 male, 2 female). The subjects were asked to seat in front of the camera at an average distance of 1.1 meters and lit from the front with ambient natural light through a window. Each subject was then instructed to perform six tasks with varying levels of head motion such as slow/fast translation between camera plane and head motion as well as small/medium head rotations. Gold-

---

[1]https://www.blender.org/

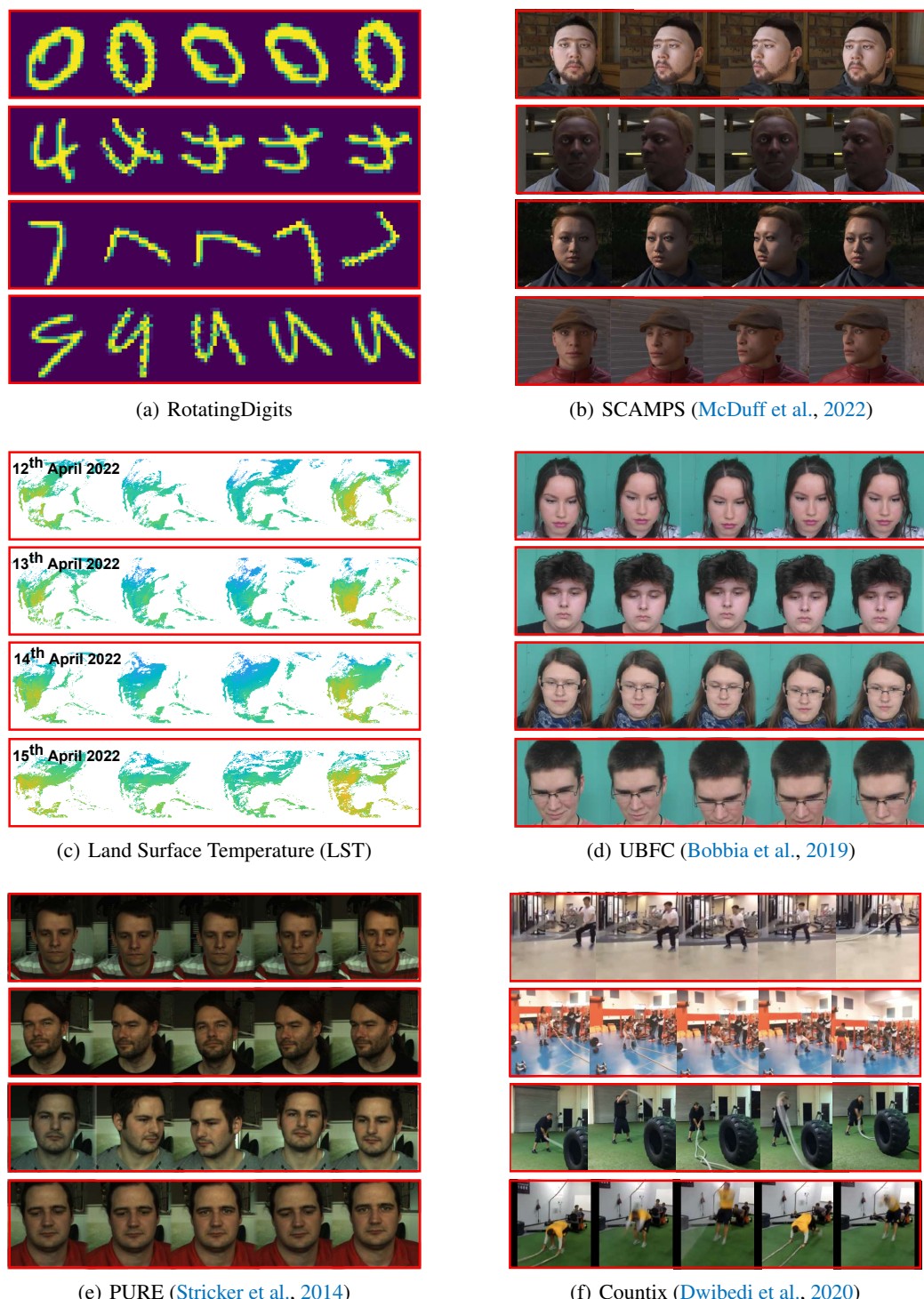

(a) RotatingDigits

(b) SCAMPS (McDuff et al., 2022)

(c) Land Surface Temperature (LST)

(d) UBFC (Bobbia et al., 2019)

(e) PURE (Stricker et al., 2014)

(f) Countix (Dwibedi et al., 2020)

Figure 7: **Examples of sequences from the datasets used in our experiments.**

standard measurements were collected with a pulse oximeter at 60Hz. The raw video resolution is $640 \times 480$. We postprocess the videos by cropping the face region and resizing them to $36 \times 36$, and downsample the ground-truth PPG signal to 30Hz from 60Hz. We manually divide each video into non-overlapping chunks (Liu et al., 2020) with a window size 180 frames (6 seconds). The resulting number of training and test samples are $1,028$ and 226, respectively.

Table 11: **Detailed statistics of the datasets used in our experiments.**

|  | Targets | Sampling freq. | Frame length | # Training set | # Val. set | # Test set |
|---|---|---|---|---|---|---|
| RotatingDigits | Rotation frequency | 30Hz | 150 | 1,000 | – | 2,000 |
| SCAMPS (McDuff et al., 2022) | Heart rate | 30Hz | 600 | 2,000 | 400 | 400 |
| UBFC (Bobbia et al., 2019) | Heart rate | 30Hz | 180 | 518 | – | 106 |
| PURE (Stricker et al., 2014) | Heart rate | 30Hz | 180 | 1,028 | – | 266 |
| Countix (Dwibedi et al., 2020) | Action counts | 20∼30Hz | 200 | 1,712 | 457 | 963 |
| LST | Temperature | Hourly | 100 | 276 | – | 92 |

**Countix** *(Action Counting)*. The Countix dataset (Dwibedi et al., 2020) is a subset of the Kinetics (Kay et al., 2017) dataset annotated with segments of repeated actions and corresponding counts. The creators crowdsourced the labels for repetition segments and counts for the selected classes. We further filter out videos that have a frame length shorter than 200, and make all videos have a fixed length of 200 frames. The resulting dataset has $1,712$ training samples, $457$ validation samples, and $963$ test samples, with a resolution of $96 \times 96$.

**Land Surface Temperature (LST)** *(Satellite Sensing)*. Land surface temperature is an indicator of the Earth surface energy budget and is widely required in applications of hydrology, meteorology and climatology. It is of fundamental importance to the net radiation budget at the Earth's surface and for monitoring the state of crops and vegetation, as well as an important indicator of both the greenhouse effect and the energy flux between the atmosphere and earth surface. We created a snapshot of data from the `NOAA GOES-16 Level 2 LST product` comprising of hourly land surfaces temperature outputs over the continental United States (CONUS). The LST measurements are sampled hourly over a 100 day period leading to $2,400$ LST maps at a resolution of $1,500 \times 2,500$. As the spatial resolution is high, we divide each map into four quarters (`North-West US`, `North-East US`, `South-West US`, and `South-East US`). We create each input sample using a window size of 100 frames with a step size of 24 (a day). The target signal is the temperature time series of the future 100 frames. The resulting dataset has 276 training samples and 92 test samples, with a spatial resolution of $100 \times 100$.

## C  EXPERIMENTAL SETTINGS

### C.1  COMPETING ALGORITHMS

We employ the following state-of-the-art SSL algorithms for comparisons.

**SimCLR** (Chen et al., 2020). SimCLR learns feature representations by contrasting images with data augmentation. The positive pairs are constructed by sampling two images with different augmentations on one instance. The negative pairs are sampled from two different images.

**MoCo v2** (He et al., 2020). MoCo learns feature representations by building large dictionaries along with a contrastive loss. MoCo maintains the dictionary as a queue of data samples by enqueuing the current mini-batch and dequeuing the oldest mini-batch. The keys are encoded by a slowly progressing encoder with a momentum moving average and the query encoder.

**BYOL** (Grill et al., 2020). BYOL leverages two neural networks to learn the feature representations: the online and target networks. The online network has an encoder, a projector, and a predictor while the target network shares the same architecture but with a different set of weights. The online network is trained by the regression targets provided by the target network.

**CVRL** (Qian et al., 2021). Contrastive Video Representation Learning (CVRL) is a self-supervised learning framework that learns spatial-temporal features representations from unlabelled videos. CVRL generates positive pairs by adding temporally consistent spatial augmentation on one videos clip and generate negative pairs by sampling two different video clips. The goal of contrastive loss is to minimize the embedding distance from the positive augmented video clips but maximize the distance from negative video clips.

## C.2 IMPLEMENTATION DETAILS

We describe the implementation details in this section. We first introduce parameters that are fixed to be the same across all methods, then detail the specific parameters for each dataset.

For all SSL methods, we follow the literature (Chen et al., 2020; He et al., 2020) and apply the same standard data augmentations in contrastive learning. For spatial augmentations, we employ `random_crop_resize`, `random_brightness`, `random_gaussian_blur`, `random_grayscale`, and `random_flip_left_right`. For temporal augmentations, we mainly employ `random_reverse` and `random_delay` (with shorter clip subsampling (Qian et al., 2021)). Unless specified, all augmentation hyper-parameters follow the original setup of each method.

**RotatingDigits.** On RotatingDigits, we adopt the network architecture as a simple 3D variant of the MNIST CNN used in (Yang et al., 2022a; Gulrajani & Lopez-Paz, 2021). In the supervised setting, we train all models for 20 epochs using the Adam optimizer (Kingma & Ba, 2014), with an initial learning rate of $10^{-3}$ and then decayed by 0.1 at the 12-th and 16-th epoch, respectively. We fix the batch size as 64 and use the checkpoint at the last epoch as the final model for evaluation. In the self-supervised setting, we train all models for 60 epochs, which ensures convergence for all tested algorithms. We again employ the Adam optimizer and decay the learning rating at the 40-th and 50-th epoch, respectively. Other training hyper-parameters remain unchanged.

**SCAMPS.** Similar to RotatingDigits, we employ the same 3D CNN architecture for all the SCAMPS experiments. In the supervised setting, we train all of the models for 30 epochs using the Adam optimizer, with an initial learning rate of $10^{-3}$ and then decayed by 0.1 at the 20-th and 25-th epoch, respectively. We fix the batch size as 32 and use the last checkpoint for final evaluation. In the self-supervised setting, we follow the same training regime in RotatingDigits as described in the previous section.

**UBFC & PURE.** Following (Liu et al., 2020; 2021), we use the temporal shift convolution attention network (TS-CAN) as our backbone model. To adapt TS-CAN on `SimPer`, we remove the attention branch and make a variant of TS-CAN which only requires 3-channel as the input instead of 6-channel. In the supervised setting, we use the Adam optimizer, learning rate of $10^{-3}$ and train the network for a total of 10 epochs. On the inner-dataset evaluation (i.e., test and validation are the same), we use last epoch from the training of 80% for the dataset and evaluate the pre-trained model on the last 20% dataset. On the cross-dataset evaluation, we use 80% of the dataset for training and 20% for checkpoint selection then evaluate the pre-trained model on a different dataset. In the self-supervised setting, all other parameters remain unchanged except that we train for 60 epochs to ensure the SSL loss converges for all algorithms.

**Countix.** We use a ResNet-3D-18 (Tran et al., 2018; He et al., 2016) architecture for all Countix experiments, which is widely used for video-based vision tasks. In the supervised setting, we train all models for 90 epochs using the Adam optimizer with an initial learning rate of $10^{-3}$ and then decayed by 0.1 at the 60-th and 80-th epoch. We fix the batch size as 32 for all experiments. In the self-supervised setting, we train all models for 200 epochs, and leave other parameters unchanged.

**LST.** Similar to Countix, we use the ResNet-3D-18 (Tran et al., 2018) network architecture for LST experiments. In the supervised setting, we train all models for 30 epochs using the Adam optimizer with a learning rate of $10^{-3}$ and a batch size of 16. In the self-supervised setting, we train all models for 60 epochs while having other hyper-parameters the same for all methods.

## C.3 EVALUATION METRICS

We describe in detail all the evaluation metrics we used in our experiments.

**MAE.** The mean absolute error (MAE) is defined as $\frac{1}{N}\sum_{i=1}^{N}|y_i - \widehat{y}_i|$, which represents the averaged absolute difference between the ground truth and predicted values over all samples.

**MAPE.** The mean absolute percentage error (MAPE) is defined as $\frac{1}{N}\sum_{i=1}^{N}|\frac{y_i-\widehat{y}_i}{y_i}|$, which assesses the averaged relative differences between the ground truth and predicted values over all samples.

**GM.** We use error Geometric Mean (**GM**) as another evaluation metric (Yang et al., 2021). GM is defined as $(\prod_{i=1}^{N}e_i)^{\frac{1}{N}}$, where $e_i \triangleq |y_i - \widehat{y}_i|$ represents the $L_1$ error of each sample. GM aims to

characterize the fairness (uniformity) of model predictions using the geometric mean instead of the arithmetic mean over the prediction errors.

**Pearson correlation** $\rho$**.** We employ Pearson correlation for performance evaluation on LST, where Pearson correlation evaluates the linear relationship between predictions and corresponding ground truth values.

# D    ADDITIONAL RESULTS AND ANALYSIS

## D.1    DATA EFFICIENCY W.R.T. REDUCED TRAINING DATA

We provide quantitative results to verify the data efficiency of `SimPer` in the presence of reduced training data. Specifically, we use SCAMPS dataset, and vary the training dataset size from $100\%$ to only $5\%$, and use it for both pre-training and fine-tuning. We show the final performance in Table 12, where `SimPer` is able to achieve consistent performance gains compared to baselines when the dataset size varies. Furthermore, the gains are more significant when the dataset size is smaller (e.g., $5\%$), demonstrating that `SimPer` is particularly robust to reduced training data.

Table 12: **Data efficiency w.r.t. reduced training data.** We vary the training dataset size of SCAMPS (size fixed for both pre-training and fine-tuning), and show the final fine-tuning performance of different methods.

| Dataset size | 100% | | 50% | | 20% | | 10% | | 5% | |
|---|---|---|---|---|---|---|---|---|---|---|
| Metrics | MAE$^\downarrow$ | MAPE$^\downarrow$ | MAE$^\downarrow$ | MAPE$^\downarrow$ | MAE$^\downarrow$ | MAPE$^\downarrow$ | MAE$^\downarrow$ | MAPE$^\downarrow$ | MAE$^\downarrow$ | MAPE$^\downarrow$ |
| SUPERVISED | 3.61 | 5.33 | 3.85 | 5.60 | 4.57 | 7.16 | 7.13 | 10.08 | 12.24 | 15.42 |
| SIMCLR (Chen et al., 2020) | 4.96 | 6.92 | 6.55 | 9.39 | 6.01 | 9.25 | 7.63 | 10.19 | 13.75 | 15.72 |
| CVRL (Qian et al., 2021) | 5.52 | 7.34 | 3.66 | 5.64 | 4.86 | 7.77 | 7.08 | 9.45 | 14.11 | 15.91 |
| **SIMPER** | **3.27** | **4.89** | **3.38** | **5.24** | **3.93** | **5.67** | **4.65** | **7.06** | **4.75** | **7.64** |
| GAINS VS. SUPERVISED | **+0.34** | **+0.44** | **+0.47** | **+0.36** | **+0.64** | **+1.49** | **+2.48** | **+3.02** | **+7.49** | **+7.78** |

## D.2    AMOUNT OF LABELED DATA FOR FINE-TUNING

We investigate the impact of the amount of labeled data for fine-tuning. Specifically, we use the whole training set of SCAMPS as the unlabeled dataset, and vary the labeled data fraction for fine-tuning. As Table 13 confirms, when the amount of labeled data is limited for fine-tuning, `SimPer` still substantially outperforms baselines by a large margin, achieving a $67\%$ relative improvement in MAE even when the labeled data fraction is only $5\%$. The results again demonstrate that `SimPer` is data efficient in terms of the amount of labeled data available.

Table 13: **Data efficiency w.r.t. amount of labeled data for fine-tuning.** We use all data from SCAMPS as unlabeled training set for self-supervised pre-training, and vary size of labeled data for fine-tuning.

| Labeled data fraction | 100% | | 50% | | 20% | | 10% | | 5% | |
|---|---|---|---|---|---|---|---|---|---|---|
| Metrics | MAE$^\downarrow$ | MAPE$^\downarrow$ | MAE$^\downarrow$ | MAPE$^\downarrow$ | MAE$^\downarrow$ | MAPE$^\downarrow$ | MAE$^\downarrow$ | MAPE$^\downarrow$ | MAE$^\downarrow$ | MAPE$^\downarrow$ |
| SUPERVISED | 3.61 | 5.33 | 3.85 | 5.60 | 4.57 | 7.16 | 7.13 | 10.08 | 12.24 | 15.42 |
| SIMCLR (Chen et al., 2020) | 4.96 | 6.92 | 4.92 | 7.09 | 5.57 | 8.46 | 7.82 | 10.53 | 13.21 | 15.64 |
| CVRL (Qian et al., 2021) | 5.52 | 7.34 | 3.79 | 5.83 | 4.83 | 7.71 | 6.82 | 9.06 | 12.18 | 13.25 |
| **SIMPER** | **3.27** | **4.89** | **3.32** | **5.13** | **3.58** | **5.44** | **3.98** | **5.81** | **4.02** | **6.27** |
| GAINS VS. SUPERVISED | **+0.34** | **+0.44** | **+0.53** | **+0.47** | **+0.99** | **+1.72** | **+3.15** | **+4.27** | **+8.22** | **+9.15** |

## D.3    ROBUSTNESS TO SPURIOUS CORRELATIONS

We provide detailed quantitative results for the spurious correlations experiment in Section 4.5. Recall that SimCLR is easy to learn information that is spuriously correlated in the training data,

Table 14: **Feature evaluation results on RotatingDigits with spurious correlations in training data.** Quantitative results in addition to Fig. 6 further verify that state-of-the-art SSL methods (e.g., SimCLR) are vulnerable to spurious correlations, and could easily learn information that is irrelevant to periodicity; In contrast, `SimPer` learns desirable periodic representations that are robust to spurious correlations.

| Metrics | FFT | | 1-NN | |
|---|---|---|---|---|
| | $\text{MAE}^{\downarrow}$ | $\text{MAPE}^{\downarrow}$ | $\text{MAE}^{\downarrow}$ | $\text{MAPE}^{\downarrow}$ |
| SIMCLR (Chen et al., 2020) | 3.06 | 125.48 | 1.49 | 80.28 |
| **SIMPER** | **0.36** | **15.04** | **0.78** | **27.03** |
| GAINS | +2.70 | +110.44 | +0.71 | +53.25 |

and the learned representations do not generalize. Table 14 further confirms the observation, where SimCLR achieves bad feature evaluation results with large MAE & MAPE errors.

In contrast, `SimPer` is able to learn the underlying frequency information even in the presence of strong spurious correlations, obtaining substantially smaller errors compared to SimCLR. The results demonstrate that `SimPer` is robust to spurious correlations, and can learn robust representations that generalize.

## D.4    ABLATION STUDIES FOR SIMPER

In this section, we perform extensive ablation studies on `SimPer` to investigate the effect of different design choices as well as its hyper-parameter stability.

### D.4.1    RANGE OF PERIODICITY-VARIANT FREQUENCY AUGMENTATION

We study the effect of using different ranges of the variant speed augmentations in `SimPer`. We use the SCAMPS dataset, and vary the speed range during `SimPer` pre-training. As Table 15 reports, using different speed ranges does not change the downstream performance by much, where all the results outperform the supervised baseline by a notable margin.

Table 15: **Ablation study on the range of speed (frequency) augmentation.** Default settings used in the main experiments for `SimPer` are marked in gray .

| SPEED RANGE | $[0.5, 1.5]$ | $[0.8, 1.8]$ | $[0.5, 2]$ | $[0.5, 3]$ | Supervised |
|---|---|---|---|---|---|
| $\text{MAPE}^{\downarrow}$ | 4.97 | 4.92 | 4.89 | 4.98 | 5.33 |

### D.4.2    NUMBER OF PERIODICITY-VARIANT AUGMENTED VIEWS

We study the effect of different number of periodicity-variant augmented views $M$ on `SimPer`. We again employ the SCAMPS dataset, and vary the number of augmented views as $M \in \{3, 5, 10, 20\}$. Table 16 shows the results, where we can observe a clear trend of decreased error rates when increasing $M$. Yet, when $M \geq 5$, the benefits of increasing $M$ gradually diminish, indicating that a moderate $M$ might be enough for the task. In the experiments of all tested datasets, to balance the efficiency while maintaining the contrastive ability, we set $M = 10$ by default.

Table 16: **Ablation study on the number of periodicity-variant augmented views.** Default settings used in the main experiments for `SimPer` are marked in gray .

| NUM. VIEWS | 3 | 5 | 10 | 20 | Supervised |
|---|---|---|---|---|---|
| $\text{MAPE}^{\downarrow}$ | 5.12 | 4.96 | 4.89 | 4.87 | 5.33 |

Table 17: **Ablation study on the choices of different periodic similarity measures.** Default settings used in the main experiments for `SimPer` are marked in `gray`.

| SIMILARITY METRICS | MXCorr | nPSD $(\cos(\cdot))$ | nPSD $(L_2)$ | Supervised |
|---|---|---|---|---|
| MAPE$^{\downarrow}$ | 4.89 | 4.88 | 4.92 | 5.33 |

### D.4.3 CHOICES OF DIFFERENT SIMILARITY METRICS

We investigate the impact of different choices of periodic similarity measures introduced in Section 3.2. Specifically, we study three concrete instantiations of periodic similarity measures: **MXCorr**, **nPSD** $(\cos(\cdot))$, and **nPSD** $(L_2)$. As Table 17 shows, `SimPer` is robust to all aforementioned periodic similarity measures, achieving similar downstream performances. The results also demonstrate the effectiveness of the proposed similarity measures in periodic learning.

### D.4.4 EFFECTIVENESS OF THE GENERALIZED CONTRASTIVE LOSS

We assess the effectiveness of the generalized contrastive loss, as compared to the classic InfoNCE contrastive loss. Table 18 highlights the results over all six datasets, where consistent gains can be obtained when using the generalized contrastive loss in `SimPer` formulation.

Table 18: **Ablation study on the effectiveness of using generalized contrastive loss in** `SimPer`**.** We show the feature evaluation results (FFT, MAE$^{\downarrow}$) with and without generalized contrastive loss across different datasets. Note that generalized contrastive loss with no continuity considered degenerates to InfoNCE (Oord et al., 2018).

| | RotatingDigits | SCAMPS | UBFC | PURE | Countix | LST |
|---|---|---|---|---|---|---|
| SimPer (InfoNCE) | **0.23** | 18.27 | 9.53 | 15.74 | 2.42 | **4.84** |
| SimPer (Generalized) | 0.22 | **14.45** | **8.78** | **13.97** | **2.06** | **4.84** |
| Gains | **+0.01** | **+3.82** | **+0.75** | **+1.77** | **+0.36** | **+0.00** |

### D.4.5 CHOICES OF DIFFERENT INPUT SEQUENCE LENGTHS

Finally, we investigate the effect of different sequence lengths on the final performance in periodic learning. To make the observations more general and comprehensive, we choose three datasets from different domains (i.e., RotatingDigits, SCAMPS, and LST) to study the effect of sequence length. We fix all the experimental setups the same as in Appendix B & C, and only vary the frame/sequence lengths with different yet reasonable choices for each dataset.

As highlighted from Table 19, the results illustrate the following interesting observations:

- For "clean" periodic learning datasets with the periodic targets being the only dominating signal (i.e., RotatingDigits), using different frame lengths do not inherently change the final result.

- For dataset with relatively high SNR (i.e., LST), `SimPer` is also robust to different frame lengths. The supervised results however are worse with shorter clips, which could be attributed to the fact that less information is used in the input.

- Interestingly, for datasets where other periodic signals might exist (i.e., SCAMPS), using shorter (but with reasonable length) videos seems to slightly improve the performance of `SimPer`. We hypothesize that for a complex task such as video-based human physiological measurement, some videos may contain multiple periodic processes (e.g., PPG, breathing, blinking, etc.). A smaller frame length may not be enough to capture some of the "slow" periodic processes (e.g., breathing), thus the features learned by `SimPer` can become even more representative for PPG or heart beats estimation. Nevertheless, the differences between various choices are still small, indicating that `SimPer` is pretty robust to different frame lengths.

Table 19: **Ablation study on the input sequence lengths.** We show the fine-tune evaluation results (MAE$^{\downarrow}$) using different yet reasonable sequence lengths across various datasets.

| | RotatingDigits | | | SCAMPS | | | LST | | |
|---|---|---|---|---|---|---|---|---|---|
| # Frames | 150 | 120 | 90 | 600 | 450 | 300 | 100 | 80 | 60 |
| SUPERVISED | 0.72 | 0.71 | 0.72 | 3.61 | 3.57 | 3.63 | 1.54 | 1.56 | 1.61 |
| **SIMPER** | **0.20** | **0.19** | **0.20** | **3.27** | **3.11** | **3.12** | **1.47** | **1.47** | **1.48** |
| GAINS | **+0.52** | **+0.52** | **+0.52** | **+0.34** | **+0.46** | **+0.51** | **+0.07** | **+0.09** | **+0.13** |

## D.5 COMPARISONS AND COMPATIBILITY WITH SOTA SUPERVISED LEARNING METHODS

As motivated in the main paper, for each specific periodic learning application, supervised learning methods (Dwibedi et al., 2020; McDuff et al., 2022; Liu et al., 2023) have achieved remarkably good results via incorporating certain domain knowledge tailored for a specific task. Therefore, we provide additional results and comparisons using SOTA algorithms on each of the tested dataset. In the following, we show existing SOTA baselines and demonstrate that SimPer could further boost the performance when jointly applied.

Table 20: **Compatibility of SimPer with SOTA supervised techniques across different datasets.** SOTA refers to RepNet (Dwibedi et al., 2020) on Countix, and refers to EfficientPhys (Liu et al., 2023) on SCAMPS, UBFC & PURE. SimPer delivers robust performance and complements the performance of SOTA models.

| | Countix | | SCAMPS | | UBFC | | PURE | |
|---|---|---|---|---|---|---|---|---|
| Metrics | MAE$^{\downarrow}$ | GM$^{\downarrow}$ | MAE$^{\downarrow}$ | MAPE$^{\downarrow}$ | MAE$^{\downarrow}$ | MAPE$^{\downarrow}$ | MAE$^{\downarrow}$ | MAPE$^{\downarrow}$ |
| SOTA | 1.03 | 0.41 | 2.42 | 4.10 | 4.14 | 3.79 | 2.87 | 2.89 |
| SIMCLR + SOTA | 1.06 | 0.43 | 2.56 | 4.17 | 4.31 | 4.02 | 2.94 | 3.25 |
| **SIMPER + SOTA** | **0.72** | **0.22** | **1.96** | **3.45** | **3.27** | **3.06** | **2.29** | **2.21** |
| GAINS | **+0.29** | **+0.19** | **+0.46** | **+0.65** | **+0.87** | **+0.73** | **+0.58** | **+0.68** |

**Countix.** In the video repetition counting domain, RepNet (Dwibedi et al., 2020), a novel neural network architecture that composed of a ResNet-50 encoder and a Transformer based predictor, is proposed to achieve advanced results for repetitious counting in the wild. We verify the compatibility of SimPer with RepNet by changing the encoder on Countix to RepNet, and compare with the vanilla supervised training as well as SimCLR. To ensure a fair and comparable setting, we train RepNet from scratch instead of using ImageNet pre-trained ResNet-50 backbones as in the original paper (Dwibedi et al., 2020).

**SCAMPS, UBFC & PURE.** In video-based human physiological sensing domain (i.e., SCAMPS, UBFC, and PURE), the main advances in the field have stemmed from better backbone architectures and network components (Liu et al., 2020; 2023; Gideon & Stent, 2021). In the main paper, for SCAMPS, since it is a synthetic dataset, we employed a simple 3D ConvNet; as for real datasets UBFC and PURE, we used a more advanced backbone model (Liu et al., 2020). To further demonstrate that SimPer can improve upon SOTA methods, we employ a recent architecture, called EfficientPhys (Liu et al., 2023), which is specialized for learning physiology from videos.

As confirmed in Table 20, when jointly applied with SOTA models, SimPer can further boost the performance and consistently achieves the best results regardless of datasets and tasks. In contrast, SimCLR is not able to improve upon SOTA supervised learning techniques. The results indicate that SimPer is orthogonal to SOTA models for learning periodic targets.

## D.6 COMPARISONS TO SSL METHODS IN HUMAN PHYSIOLOGICAL MEASUREMENT

In video-based human physiological measurement domain, recent works (Wang et al., 2022; Gideon & Stent, 2021) have proposed to leverage contrastive SSL for better learned features and downstream performance in the corresponding application (e.g., heart rate estimation). They studied specific SSL

Table 21: **Comparisons between `SimPer` and additional SSL baselines on human physiological measurement datasets.** Compared to customized SSL algorithms in the specific domain, `SimPer` still delivers robust performance and consistently achieves the best results.

| Metrics | SCAMPS | | UBFC | | PURE | |
|---|---|---|---|---|---|---|
| | MAE$^\downarrow$ | MAPE$^\downarrow$ | MAE$^\downarrow$ | MAPE$^\downarrow$ | MAE$^\downarrow$ | MAPE$^\downarrow$ |
| *Without face saliency module:* | | | | | | |
| (Gideon & Stent, 2021) | 3.53 | 5.26 | 4.98 | 4.61 | 4.18 | 4.70 |
| (Wang et al., 2022) | 3.71 | 5.54 | 5.07 | 4.88 | 4.32 | 4.95 |
| **SIMPER** | **3.27** | **4.89** | **4.24** | **3.97** | **3.89** | **4.01** |
| *With face saliency module:* | | | | | | |
| (Gideon & Stent, 2021) | 3.51 | 5.15 | 4.88 | 4.29 | 4.03 | 4.28 |
| (Wang et al., 2022) | 3.61 | 5.40 | 5.02 | 4.86 | 4.07 | 4.33 |
| **SIMPER** | **2.94** | **4.35** | **4.01** | **3.68** | **3.47** | **3.76** |

Table 22: **Comparisons between `SimPer` and additional SSL baselines on general periodic learning datasets other than human physiological measurement ones.** When extending to general periodic learning tasks, SSL baselines tailored for human physiological measurement (Wang et al., 2022; Gideon & Stent, 2021) no longer provide benefits, and sometimes perform even *worse* than the vanilla supervised learning. In contrast, `SimPer` consistently and substantially exhibits strengths in general periodic learning across all domains.

| Metrics | RotatingDigits | | Countix | | LST | |
|---|---|---|---|---|---|---|
| | MAE$^\downarrow$ | MAPE$^\downarrow$ | MAE$^\downarrow$ | GM$^\downarrow$ | MAE$^\downarrow$ | $\rho^\uparrow$ |
| SUPERVISED | 0.72 | 28.96 | 1.50 | 0.73 | 1.54 | **0.96** |
| (Gideon & Stent, 2021) | 0.70 | 28.03 | 1.58 | 0.81 | 1.62 | 0.92 |
| (Wang et al., 2022) | 0.77 | 29.44 | 1.68 | 0.94 | 1.64 | 0.89 |
| **SIMPER** | **0.20** | **14.33** | **1.33** | **0.59** | **1.47** | **0.96** |

methods tailored for video-based human physiological measurement, and as a result, many of the proposed techniques therein only apply to that specific domain (e.g., the face detector, the saliency sampler, and the strong assumptions that are derived from the application context, cf. Table 1 in (Gideon & Stent, 2021)). Nevertheless, it is possible to extend the SSL objectives therein to other general periodic learning domains. In this section, we provide additional experimental results and further discussions, which distinguish `SimPer` from these prior works.

**Comparisons on the human physiological measurement task.** We first compare `SimPer` against the aforementioned SSL methods (Gideon & Stent, 2021; Wang et al., 2022) on the human physiological measurement task. To provide a fair comparison, we fix all methods to use a simple 3D ConvNet backbone (Gulrajani & Lopez-Paz, 2021) on SCAMPS, and a TS-CAN backbone (Liu et al., 2020) on UBFC and PURE as stated in Appendix C. As Table 21 demonstrates, `SimPer` outperforms these SSL baselines across all tested human physiology datasets by a notable margin. We break the results out to confirm that they hold regardless of whether we include the customized face saliency module (Gideon & Stent, 2021) or not.

**Comparisons on other periodic learning tasks.** We further extend the comparisons to other general periodic learning tasks. We directly apply the SSL objectives in (Gideon & Stent, 2021; Wang et al., 2022) to other domains and datasets involving periodic learning, and show the corresponding results in Table 22. The table clearly shows that the SSL objectives in the referenced papers do not provide a benefit in other periodic learning domains, and sometimes perform even *worse* than the vanilla supervised baseline. The above results further emphasize the significance of `SimPer`, which consistently and substantially exhibits strengths in general periodic learning across all domains.

### D.7 VISUALIZATION OF LEARNED FEATURES

Since representations learned in periodic data naturally preserves the periodicity information, we can directly plot the learned 1-D features for visualization. Fig. 8 shows the learned feature comparison

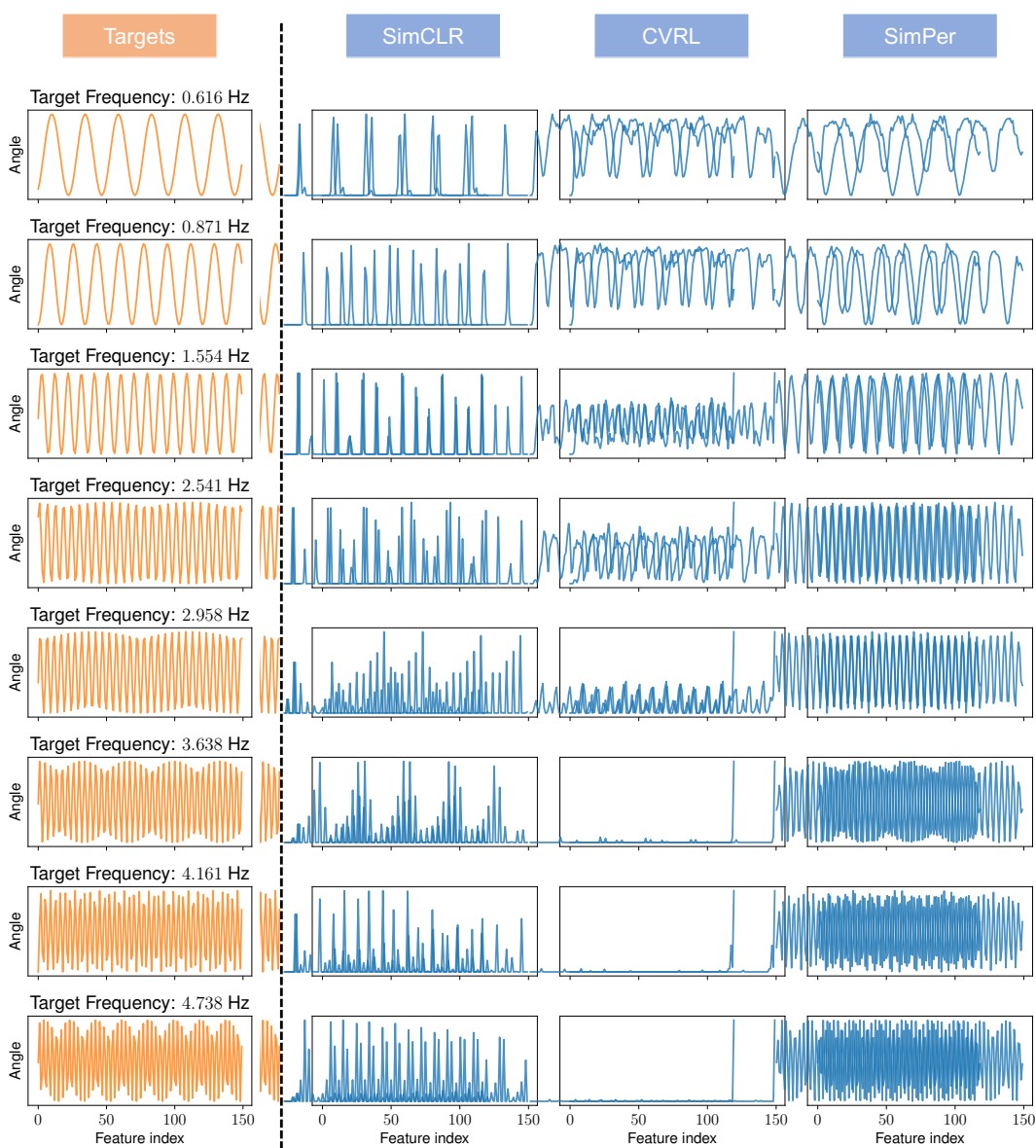

Figure 8: **Visualization of learned periodic representations.** We directly plot the 1-D feature vector of data in the test set of RotatingDigits with different underlying target frequencies (left), via different self-supervised learning methods (right). Existing SSL solutions (Qian et al., 2021; Chen et al., 2020) fail to learn meaningful periodic representations, whereas `SimPer` is able to capture the underlying periodicity information.

between SimCLR, CVRL, and `SimPer`, together with the underlying periodic information (rotation angle & frequency) in RotatingDigits. As the figure verifies, `SimPer` consistently learns the periodic information with different frequency targets, delivering meaningful periodic representations that are robust and interpretable. In contrast, existing SSL methods cannot capture the underlying periodicity, and fail to learn useful representations for periodic learning tasks.

# E    BROADER IMPACTS AND LIMITATIONS

**Limitations.** There are some limitations to our approach in its current form. The `SimPer` features learnt in some cases were not highly effective without certain fine-tuning on a downstream task. This may be explained by the fact that some videos may contain multiple periodic processes (e.g.,

pulse/PPG, breathing, blinking, etc.). A pure SSL approach will learn features related to all these periodic signals, but not information that is specific to any one. One practical solution for this limitation could be incorporating the *frequency priors* of the targets of interest. Precisely, one can filter out unrelated frequencies during `SimPer` pre-training to force the network to learn features that are constrained within a certain frequency range. We leave this part as future work.

**Broader Impacts.** While our methods are generic to tasks that involve learning periodic signals, we have selected some specific tasks on which to demonstrate their efficacy more concretely. The measurement of health information from videos has tremendous potential for positive impact, helping to lower the barrier to access to frequent measurement and reduce the discomfort or inconvenience caused by wearable devices. However, there is the potential for negative applications of such technology. Whether by negligence, or bad intention, unobtrusive measurement could be used to measure information covertly and without the consent of a user. Such an application would be unethical and would also violate laws in many parts of the world[2]. It is important that the same stringent measures applied to traditional medical sensing are also applied to video-based methods. We will be releasing code for our approach under a Responsible AI License (RAIL) (Contractor et al., 2022) to help practically mitigate unintended negative behavioral uses of the technology while still making the code available.

---

[2]https://www.ilga.gov/legislation/ilcs/ilcs3.asp?ActID=3004&ChapterID=57

