# OpenReview forum: "SimPer: Simple Self-Supervised Learning of Periodic Targets"
_ICLR.cc/2023/Conference — ICLR 2023 notable top 5%_

### Official Review · Reviewer_y54W · 2022-10-22

**Confidence:** 4
**Correctness:** 3
**Technical Novelty And Significance:** 3
**Empirical Novelty And Significance:** 4
**Recommendation:** 8

**Clarity, Quality, Novelty And Reproducibility:**

The work has high quality and the writing and exposition are very clear. There are a few additional ablations that could be helpful (see weaknesses), but overall the experiments are very comprehensive and supportive of the claims made in the abstract and introduction.

I cannot fully assess the novelty of the paper regarding the task of periodicity estimation/representation. The related work section covers previous work on this problem and it is consistent with my literature scan.

There should be no major issues with respect to reproducibility, as the method is explained very well and pseudocode is provided in the appendix. It would certainly be helpful if the authors also provided the code to reproduce the experiments.

**Strength And Weaknesses:**

**Strengths**

- The overall idea and the individual design choices are well-motivated and explained very clearly in the text and supported by suitable illustrations.
- It is transparently stated that the method makes use of an inductive bias to construct suitable augmentations and loss functions.
- The empirical analysis is very comprehensive. It clearly shows the benefits of the proposed method and provides helpful ablations.
- The writing and exposition of the paper are very clear.


**Weaknesses**

- It is not sufficiently clear how to choose the sequence length and how important the hyperparameter might be for the encoding of periodic information. For instance, a very long sequence might contain multiple subsequences that exhibit different periodicities. I think it would be helpful to include an ablation for the sequence length.
- Do I correctly understand that the baselines SimCLR, MoCo, and BYOL use individual frames as inputs, whereas CVRL and SimPer use sequences as inputs? In that case, it would not be surprising that the frame-based approaches do not encode periodic information, which would be difficult to estimate from a single frame.
- It would be interesting to measure not only the periodic information that is encoded, but also how much other information (object class, etc; depending on the dataset) is retained in the encoding. For instance, Figure 6 shows that the spurious feature (digit appearance) is encoded relatively well for the training data, even though it is not an invariant feature that would help discriminate between positive and negative pairs.

**Minor comments and questions:**
- In Figure 4a, why do you plot fewer data points when you decrease the training dataset size? Are the learned representations not based on a test dataset of a fixed size?
- Why is the performance of the proposed method on the LST dataset not much better compared to the baselines?
- In the context of the Nyquist sampling theorem, it would be helpful if you briefly stated its results and why it is relevant.
- Did you decrease the margins for the abstract?


**Summary Of The Paper:**

The paper proposes a contrastive learning framework for encoding periodic information from high-dimensional input sequences. The proposed method (SimPer) leverages an inductive bias present in most periodic processes to define suitable data augmentations, feature similarity measures, and loss functions. To create negative pairs, the method intervenes on a sequence by changing its frequency (by changing its speed). The proposed similarity metrics take into account temporal offsets or the power spectrum density of the signals using existing measures. The loss function is an interesting generalization of the InfoNCE loss where the loss for a pair of instances is weighted using side information on the similarity of the given pair (e.g., the absolute difference in the speedup of two sequences). A thorough empirical analysis shows that SimPer encodes periodic information well – significantly better than frame-based methods (SimCLR, MoCo, BYOL) and one approach based on sequences (CVRL). SimPer shows improved data efficiency compared to self-supervised and supervised methods, and improved accuracy for fine-tuning, transfer learning, OOD generalization, and robustness to spurious correlations compared to other self-supervised methods.

**Summary Of The Review:**

The paper presents a simple framework that is well-motivated and quite intuitive. The paper has an empirical focus and, appropriately, provides comprehensive experiments and ablations. One exception is that the paper could provide more information regarding the choice of sequence length, which might be an important hyperparameter. Other than that, I have no major concerns and recommend acceptance.

---

> ### Author Response · Authors · 2022-11-17
> **Response to Reviewer y54W (3/3)**
>
> > *In the context of the Nyquist sampling theorem, it would be helpful if you briefly stated its results and why it is relevant.*
>
> Thanks for your comment, and we apologize if it was not clear in the original text. Since the principle of periodicity-variant augmentations is to alter the frequency by changing the speed of input sequences, the underlying periodic target signals would also be altered. Specifically:
>
> 1. When the speed is being ***reduced***, we are effectively interpolating over the input sequences, which in principle does not have constraints and is not related to the Nyquist sampling theorem;
> 2. When the speed is being ***increased***, we are effectively downsampling the video (albeit continuously and not simply dropping frames). According to the Nyquist theorem, the maximum frequency we could determine from a video is half that of the sampling rate of the given video, which constrains how much we can alter the video. Therefore, one needs to be careful not to downsample the video too aggressively.
>
> &nbsp;
>
> > *Did you decrease the margins for the abstract?*
>
> Thank you for pointing this out. We did not intentionally decrease the margins of the abstract. After some investigations, it turns out that the ICLR template has some conflicts with our custom `itemize` environment, which caused the issue. We have fixed the margins in the newly revised version. (Abstract wording is rephrased a bit to fit the 9-page limit.)

---

> ### Author Response · Authors · 2022-11-17
> **Response to Reviewer y54W (2/3)**
>
> > *It would be interesting to measure not only the periodic information that is encoded, but also how much other information (object class, etc; depending on the dataset) is retained in the encoding. For instance, Figure 6 shows that the spurious feature (digit appearance) is encoded relatively well for the training data, even though it is not an invariant feature that would help discriminate between positive and negative pairs.*
>
> Thank you for this suggestion. We would like to first make some clarifications regarding your comments on Figure 6. For SimPer, the spurious feature (digit appearance) is actually **not** encoded or learned: since for the training data, we explicitly make the target frequency spuriously correlated with digit appearances in the training set (cf. Section 4.5, last sentence in the first paragraph), as long as SimPer learns the underlying frequency well, it will by definition induce a separated digit representation for the training set. Moreover, since it does not learn the spurious feature, in the test set where the correspondence is removed between digits and frequencies, SimPer still exhibits good features (if it had learned the digit appearance the clustering would also be separated by digit appearance in the test set).
>
> On a related note, we believe that these experiments demonstrate that SimPer helps to suppress other unrelated information (e.g., object category) and preserve only the periodicity-related information in the data. This is largely due to the fact that we maximally exploit the periodic (frequency) inductive bias in SimPer’s design. As discussed in the last paragraph of Section 3.1, the temporal **self**-contrastive scheme naturally leads to hard negative samples, as *periodic information* is directly being contrasted, while other information (e.g., frame appearance) are maximally preserved across negative samples. Since we introduce negative samples directly from the same anchor sample (which means information such as frame appearance is preserved), contrasting over such negative samples would make the model not learn such information.
>
> Nevertheless, we do agree that certain information beyond periodicity would be potentially beneficial to certain tasks, and definitely is an interesting future direction. In the current work, it is not immediately obvious that one could easily define or categorize such scenarios in a principled manner.  We hope and believe that this work would motivate future investigation into these questions.
>
> &nbsp;
>
> > *In Figure 4a, why do you plot fewer data points when you decrease the training dataset size? Are the learned representations not based on a test dataset of a fixed size?*
>
> To clarify, the plot is actually made on the training set, which answers the question why there are fewer data points. Since in this case, the training and test sets are I.I.D. with the same data distribution, the visualization on the test set will be in principle the same as the training one, except with more data points (as we do have a test set of a fixed size). We plot the training set just to make the visualization more interpretable and coherent with the design intention, that learned representations can become worse with fewer data points for supervised / current SSL methods, but not for SimPer.
>
> &nbsp;
>
> > *Why is the performance of the proposed method on the LST dataset not much better compared to the baselines?*
>
> Thank you for your point, and this is a great question. The performance difference could be attributed to the specific task. Unlike other datasets or tasks where the goal is to extract the underlying periodic signals from a given input sequences, the task in the LST context is to model and predict **future** temperature time series from current satellite inputs (see Appendix B). In this case, good periodic representations would definitely be helpful for future trend prediction, but may not need to be necessarily enforced when training data is already sufficient (i.e., the goal is to learn the mapping from now to future, rather than extracting the periodic targets directly).
>
> To further assess this argument as well as understanding the behavior of SimPer, we repeat the data efficiency experiment by reducing the training set size, but this time on the LST dataset. We present the corresponding results in the following table (MAE, $\downarrow$):
>
> | |100% |50%|20%|10%|5% |
> | - | :-: | :-: | :-: | :-: | :-: |
> | Supervised |1.54 |1.78 |2.18 |2.79 |3.56 |
> |SimCLR |1.54 |1.77 |2.13 |2.76 |3.42|
> |SimPer |**1.47**|**1.56**|**1.78**|**1.96**|**2.21**|
>
> As the table highlights, when the data becomes insufficient to directly learn the mapping, the performance gains of SimPer become larger and substantial compared to baselines, indicating that learning periodic representations via SimPer is still beneficial.

---

> ### Author Response · Authors · 2022-11-17
> **Response to Reviewer y54W (1/3)**
>
> Thank you very much for your supportive remarks and constructive feedback! We are delighted to see that you found the overall idea and design choices well-motivated, and that the paper is presented and explained very clearly. Below, we provide additional results and clarifications, which we hope will encourage the reviewer to further increase the score.
>
> &nbsp;
>
> > *It is not sufficiently clear how to choose the sequence length and how important the hyperparameter might be for the encoding of periodic information. For instance, a very long sequence might contain multiple subsequences that exhibit different periodicities. I think it would be helpful to include an ablation for the sequence length.*
>
> Thank you for your great point! We agree with you that sequence length is an interesting yet important parameter to study, and corresponding results would be useful for us to provide a more informed context of our work. To make the observations more general and comprehensive, we choose three datasets from different domains, i.e., RotatingDigits, SCAMPS, and LST to study the effect of sequence length (for Countix it is not as obvious how we might modify the length as the associate counts are fixed). For clarity we fix all the experimental setups as in Appendix B & C, and only vary the frame/sequence lengths with different (but reasonable) choices.
>
> | | RotatingDigits | | | SCAMPS| | | LST| | |
> | - | :-: | :-: | :-: | :-: | :-: | :-: | :-: | :-: | :-: |
> |# Frames |150 |120 |90 |600 |450 |300| 100| 80 |60|
> |Supervised |0.72 |0.71 |0.72 |3.61 |3.57 |3.63 |1.54 |1.56 |1.61|
> |SimPer|**0.20**|**0.19**|**0.20**|**3.27**|**3.11**|**3.12**|**1.47**|**1.47**|**1.48**|
>
> As highlighted from the table above, the results illustrate the following interesting observations:
> 1. For “clean” periodic learning datasets with the periodic targets being the only dominating signal (i.e., RotatingDigits), using different frame lengths do not inherently change the final performance;
> 2. For dataset with relatively high SNR (i.e., LST, where the periodicity is still the dominating signal due to regular earth movement), SimPer is also robust to different frame lengths. The supervised results however are worse with shorter clips, which could be attributed to the fact that less information is used in the input.
> 3. Interestingly, for datasets where other periodic signals might exist such as SCAMPS (as the reviewer also suggested), using shorter (but with reasonable length) videos seems to slightly improve the performance of SimPer. We hypothesize that for a complex task such as video-based human physiological measurement, some videos may contain multiple periodic processes (e.g., pulse/PPG, breathing, blinking, etc.). A smaller frame length may not be enough to capture some of the “slow” periodic processes (e.g., breathing), thus the features learned by SimPer can become even more representative for PPG / heart beats estimation. Nevertheless, the differences between various choices are still not very large, indicating that SimPer is pretty robust to frame length choices. We really appreciate this suggestion and believe this additional analysis will be very informative to the readers of our paper.
>
> To summarize, with a reasonable range of frame lengths, SimPer tends to be consistent and robust to the sequence length choices across datasets, and the performance is not dramatically affected by changing the length. We have added the above results as well as a more elaborate discussion in **Section 4.6, Appendix D.4.5, and Table 19** to address your concerns about different choices of frame lengths.
>
> &nbsp;
>
> > *Do I correctly understand that the baselines SimCLR, MoCo, and BYOL use individual frames as inputs, whereas CVRL and SimPer use sequences as inputs? In that case, it would not be surprising that the frame-based approaches do not encode periodic information, which would be difficult to estimate from a single frame.*
>
> To clarify, all SSL methods (including SimCLR, MoCo, and BYOL) use *sequences* as inputs; that being said, the inputs of all SSL methods are kept the **same** throughout all experiments. Since we use the same backbone to extract intermediate features, the resulting feature shape is $\mathbb{R}^{T}$ for all experiments, with $T$ being the sequence length. SSL methods like SimCLR or CVRL then operate over these intermediate features with their corresponding objectives. We also note that the setting can be easily extended to high-dimensional features (in addition to the time dimension) by simply averaging across other dimensions (cf. Discussions in main paper, page 4, “Concrete Instantiations”).
>
> To summarize, all SSL methods are fixed to use the same sequences as inputs. This ensures that the comparisons are fair and controlled, which also demonstrates that the benefits of SimPer stem from the algorithmic improvements.

---

> ### Comment · Reviewer_y54W · 2022-11-22
> **Response to the authors**
>
> Thank you for the detailed answers and additional experiments. The answers are very helpful and address my concerns. I tend to keep my recommendation at 8.

---

> > ### Author Response · Authors · 2022-11-23
> > **Glad to learn that all your concerns are addressed! Note on additional (related) experiments**
> >
> > Thank you for your positive feedback! We are delighted to see that our responses have addressed all your concerns.
> >
> > On a related note, we would like to point you to the latest experiments we performed w.r.t. the time window lengths during evaluation (see [our latest response to Reviewer DG5c](https://openreview.net/forum?id=EKpMeEV0hOo&noteId=-4H8xxJP_CB)). This is similar to your previous suggestion on the input sequence length, but differs from the fact that it focuses on the evaluation protocol (i.e., more fine-grained evaluation with shorter length). We confirm that using different segment lengths does not change the evaluation results, where SimPer consistently outperforms the baselines. This highlights and signifies that the evaluation protocols are reasonable and pretty robust.
> >
> > We thank the reviewer again for your time and insightful feedback. Please feel free to let us know if there are other clarifications or experiments we can offer.

---

### Official Review · Reviewer_DG5c · 2022-10-24

**Confidence:** 2
**Correctness:** 4
**Technical Novelty And Significance:** 3
**Empirical Novelty And Significance:** 3
**Recommendation:** 8

**Clarity, Quality, Novelty And Reproducibility:**

* The paper is well written and easy to follow

* The quality of the submission is high. The figures and tables are placed neatly to improve the readability of paper.

* The idea of using simple augmentation techniques for temporal learning is novel.

* The paper contains reasonable amount of information that could help in reproducibility.

**Strength And Weaknesses:**

**Strengths**

* The method improves results over self-supervised methods such as SimCLR.

* The paper is well written and easy to follow.

* The paper advances simple augmentation technique for temporal contrastive learning framework.

**Weaknesses**

* Although the approach is interesting, the experimental section would require additional results to benchmark thoroughly against CVLR. For example, this submission does not include results on Kinetics-400. (as opposed to a subset)

* There has already been self-supervised learning approaches in the area of human physiology. This submission does not compare with this ICCV paper. [ref 1]. This paper already explores the idea of maximal cross correlation. (MCC)

* It is not clear if the evaluation protocol is good enough for representations. Especially, for computing a single HR value over a large window. In other words, even if the representations are very bad for large portion of the video, the evaluation protocol would not be able to detect it.

* Additional comparisons with baselines: The results on known traditional methods are better on SCAMPS dataset than the proposed method. These baseline methods have not been to be included in this paper. It would be beneficial to the reader to have a comparison in the paper.

**References**

[ref 1] The Way to my Heart is through Contrastive Learning: Remote Photoplethysmography from Unlabelled Video, ICCV'21
[ref 2] SCAMPS: Synthetics for Camera Measurement of Physiological Signals, NeurIPS'22



**Summary Of The Paper:**

SimPer is a self-supervised learning method for learning representations from periodic data such human physiology. In this paper the authors demonstrate that existing baselines such as SimCLR do not work well. The key idea in this paper is to revise the idea of feature similarity for temporal data. More specifically, the idea is to improve similarity of temporally close features (defined through index-shifting and reversing augmentations). A "random" feature is defined to be a negative feature. The results demonstrate that SimPer is performing better than some of the self-supervised learning methods.

**Summary Of The Review:**

Comparing the strengths and weaknesses, this paper can be further improved to study why existing unsupervised methods are superior to the proposed method. Hence, in my opinion, the paper is not ready for acceptance.

---

> ### Author Response · Authors · 2022-11-17
> **Response to Reviewer DG5c (4/4)**
>
> > *Comparing the strengths and weaknesses, this paper can be further improved to study why existing unsupervised methods are superior to the proposed method.*
>
> We did not fully understand this comment. We guess that the reviewer probably meant *“why SimPer is superior to existing unsupervised methods”* rather than “is not”?
>
> We hope the clarifications above and the contributions recognized by [Reviewer pX9g](https://openreview.net/forum?id=EKpMeEV0hOo&noteId=_0md3jBaKf) and [Reviewer y54W](https://openreview.net/forum?id=EKpMeEV0hOo&noteId=-uBvd5KXlV) highlight why we argue SimPer is superior to existing methods. In order to avoid any confusion please let us briefly summarize a few integral points:
>
> **(1) The design of SimPer is *intrinsically* different to other SSL methods.**
>
> SimPer achieves substantially better results than existing SSL methods because it “leverages an **inductive bias** to construct suitable *augmentations* and *loss functions*” ([Reviewer y54W](https://openreview.net/forum?id=EKpMeEV0hOo&noteId=-uBvd5KXlV)), and “the algorithm is well designed for periodic signals to maximally extract useful information” ([Reviewer pX9g](https://openreview.net/forum?id=EKpMeEV0hOo&noteId=_0md3jBaKf)).
>
> We would like to again emphasize the design philosophy of SimPer:
> 1. *How to define positive and negative in periodic learning? (**Section 3.1**)*: Periodicity-variant & periodicity-invariant augmentations w.r.t. the **same** sample;
> 2. *How to define feature similarity in periodic learning? (**Section 3.2**)*: Periodic feature similarity, a **family** of feature similarity measure;
> 3. *How to enable contrastive learning over continuous targets? (**Section 3.3**)*: The generalized InfoNCE loss.
>
> All related ablation studies on these design choices are supported by experiments (see Appendix D.4.1–D.4.4), demonstrating that the design choices make SimPer better than existing SSL methods for learning periodic targets. (Other methods did not explore the intrinsic properties of periodic data learning.)
>
> **(2) The thorough evaluation results, supported by both *quantitative* and *qualitative* measures, further elucidate why SimPer is better.**
>
> 1. The UMAP visualizations of the learned feature space confirm that current SSL methods cannot learn meaningful periodic representations, and/or the learned features exhibit low frequency resolution (Figure 1, 4, 5)
> 2. The direct plot of the learned 1D features helps us assess the feature quality compared with the ground truth (Figure 8, Appendix D.7). These plots clearly show that existing SSL schemes (e.g., SimCLR, CVRL) fail to learn meaningful periodic representations.
>
> **(3) Beyond accuracy: Benefits in low-shot learning, transfer learning, zero-shot generalization, robustness to spurious correlations.**
>
> Finally, as highlighted by both [Reviewer pX9g](https://openreview.net/forum?id=EKpMeEV0hOo&noteId=_0md3jBaKf) and [Reviewer y54W](https://openreview.net/forum?id=EKpMeEV0hOo&noteId=-uBvd5KXlV), we did comprehensive comparisons between SimPer and other SSL methods in the following regards: **few-shot learning** (Figure 4, Table 12 & 13), **transfer learning** (Table 9), **OOD generalization** (Figure 5, Table 10), and **robustness to spurious correlations** (Figure 6, Table 14). These topics cover the *robustness*, *data efficiency*, and *generalization* of the SSL algorithms. Through extensive explorations and comparisons, we believe the advantages of SimPer over other SSL methods in periodic learning are well understood and presented.
>
>
> We are more than happy to discuss and answer more questions if you have any remaining concerns. If the above discussions and results successfully addressed all your concerns, please consider raising your summary score.
>
> &nbsp;
>
> ---
> References:
> 1. Dwibedi et al. Counting out time: Class agnostic video repetition counting in the wild. CVPR 2020.
> 2. Kay et al. The Kinetics Human Action Video Dataset. 2017.
> 3. Welch. Lower bounds on the maximum cross correlation of signals. IEEE Transactions on Information theory, 20(3):397–399, 1974.
> 4. McDuff. Camera measurement of physiological vital signs. ACM Computing Surveys (CSUR), 2021.
> 5. He et al. Momentum contrast for unsupervised visual representation learning. CVPR 2020.
> 6. Chen et al. A simple framework for contrastive learning of visual representations. ICML 2020.
> 7. Qian et al. Spatiotemporal contrastive video representation learning. CVPR 2021.
> 8. Gideon et al. The Way to my Heart is through Contrastive Learning: Remote Photoplethysmography from Unlabelled Video. ICCV 2021.
> 9. Liu et al. Multi-task temporal shift attention networks for on-device contactless vitals measurement. NeurIPS 2020.
> 10. Liu et al. EfficientPhys: Enabling simple, fast and accurate camera-based vitals measurement. WACV 2023.

---

> > ### Comment · Reviewer_DG5c · 2022-11-18
> > **Response**
> >
> > I highly appreciate clarification on the several aspects of paper. These new results are indeed are encouraging and I want to sincerely thank authors for including them in the Appendix. I understand the importance of your work in itself and understand why it is not required to perform well on specific downstream tasks. However, I continue to believe that a framework designed for self-supervised temporal learning ought to be able to perform better on fine-grained physiology tasks since this requires the approach to more fundamentally understand the dynamics of fine-grain changes - which is are more difficult to learn that stable long term periodic signal. I understand that authors argue that the evaluation protocols are empirically correlated which is not a strong indicator of performance, in my opinion.
> >
> > However, I am satisfied with the response provided by authors to my review and other reviewers and am increasing the points to a weak accept.

---

> > > ### Author Response · Authors · 2022-11-23
> > > **SimPer *DOES* perform better on physiology tasks. Fine-grained results & visualizations further support our claims (2/2)**
> > >
> > > ***2. Error distribution when using short video segments for evaluation***
> > >
> > > We go a step further to visualize the error distributions when using small segments for evaluation. If the reviewer’s claim, that the “*representations are very bad for large portion of the video*” was true, we would expect that many segments would have large errors, and the error distribution would not be skewed heavily toward 0.
> > >
> > > To this end, we provide the visualization results in [**[this anonymous link](https://drive.google.com/file/d/12khhrJwxDS-AbextB52Ay7HBeEZPLCo_/view?usp=sharing)**] for PURE and UBFC with different time window lengths.
> > >
> > > As the figure demonstrates, the error distributions when using short time windows are ***actually concentrated*** in the low-error region, and the number of segments with larger errors are very small. In our experience, this is typical for the task of heart rate estimation from video (for example see Fig. 6 in [1]). This indicates that the number of outliers is actually ***small***, and the learned representations are *consistent across different locations within the video*. Furthermore, the results hold across different **segment lengths** and **datasets**, showing that the issue raised by the reviewer (“*representations are very bad for large portion of the video*”) ***does not exist*** in real human physiology datasets.
> > >
> > > &nbsp;
> > >
> > > **To summarize**, whether we use shorter time windows for evaluation or not, our results are generally unaffected, and SimPer consistently outperforms the baselines in fine-grained evaluation scenarios. The results also highlight that the current evaluation protocols are indeed robust and consistent, and are able to determine whether the learned periodic representations are good or not.
> > >
> > > &nbsp;
> > >
> > > We believe the above results and clarifications should have addressed all your concerns, and made you more confident about the novelty, significance, and completeness of our paper. We thank you again for your time and feedback. We hope you can re-evaluate the paper based on our clarifications, and would really appreciate it if you can consider updating the reviews, and raising your score accordingly.
> > >
> > > &nbsp;
> > >
> > > ---
> > > Reference link:
> > > 1. Estepp, J. R., Blackford, E. B., & Meier, C. M. Recovering pulse rate during motion artifact with a multi-imager array for non-contact imaging photoplethysmography. IEEE international conference on systems, man, and cybernetics (SMC). 2014.
> > > 2. (***Anonymous figure***) Error distribution w.r.t. different evaluation segment lengths: https://drive.google.com/file/d/12khhrJwxDS-AbextB52Ay7HBeEZPLCo_/view?usp=sharing

---

> > > ### Author Response · Authors · 2022-11-23
> > > **SimPer *DOES* perform better on physiology tasks. Fine-grained results & visualizations further support our claims (1/2)**
> > >
> > > Thanks for your feedback! We are glad to see that our clarifications successfully resolved most of your concerns, and that the additional results are encouraging and strengthen the evidence for our approach. We believe that you have one remaining comment about “*whether the evaluation for the human physiology sensing task is good or not*”. To further clarify this point, we would like to provide additional evidence, experiments and clarifications:
> > >
> > > **First**, we would like to address your point that SimPer: “*it is not required to perform well on specific downstream tasks*”. In fact, throughout all of the experiments, SimPer ***does*** perform well on all downstream tasks. Especially for the human physiology measurement task, we have compared with:
> > > - SOTA general (contrastive) SSL methods (**Tables 3, 4, 5, 8, 9, 12, 13, 14**)
> > > - SOTA specific SSL methods for human physiology measurement (**Tables 21, 22**, **Appendix D.6**)
> > > - SOTA supervised learning methods (**Table 20**, **Appendix D.5**)
> > >
> > > All the above results confirm and highlight that SimPer does offer performance benefits on all downstream tasks.
> > >
> > > **Second**, we would like to re-emphasize that the video-level HR evaluation protocol is a standard procedure. Almost all the literature and recent papers in the video human physiology measurement domain, including the references that the reviewer mentioned, use a similar evaluation protocol, specifically an FFT applied to the output to estimate a video-level HR label. We would argue that this is a pretty widely adopted form of evaluation in the human physiology measurement field.
> > >
> > > **Nevertheless**, we agree that reporting results over shorter time windows is very interesting and a more challenging bar against which to compare. To directly address this point in a rigorous way, we performed the following experiments:
> > > - We show that even when evaluating over short time **segments** (i.e., fine-grained evaluation), SimPer still ***substantially outperforms*** the baselines, and that using different segment lengths for evaluation ***does not*** change the evaluation results in principle.
> > > - We show that the error distributions when using short segments for evaluation are ***indeed concentrated*** in the low-error region across different segment lengths and datasets, demonstrating that the issue raised by the reviewer (“*representations are very bad for large portions of the video*”) is not a problem here.
> > >
> > > &nbsp;
> > >
> > > ***1. Fine-grained evaluation results on human physiology datasets***
> > >
> > > To begin with, we use the real human physiology datasets (i.e., UBFC & PURE) for fine-grained evaluation, per the reviewer’s request. In addition to using the full video (typically 30-60 seconds), we use ***sliding windows*** with different window lengths, and evaluate the average mean absolute error (MAE, $\downarrow$) in heart rate computed via an FFT over each of the windows. We present the results below. We use window lengths of $\\{  5s, 10s, 15s, 20s \\}$, and use a stride of $1s$ to iterate over all segments.
> > >
> > > | |UBFC| | | | |PURE| | | | |
> > > |-|-|-|-|-|-|-|-|-|-|-|
> > > |Window/segment length (seconds)| 5| 10| 15| 20| full| 5| 10 |15| 20 |full|
> > > |Supervised|6.10|5.96|5.86|5.57|5.13|4.67|4.65|4.33|4.29|4.25|
> > > |SimCLR|6.40|6.33|6.18|5.85|5.32|5.25|5.21|4.98|4.91|4.86|
> > > |SimPer|**4.45**|**4.39**|**4.41**|**4.32**|**4.24**|**4.02**|**4.03**|**3.96**|**3.88**|**3.89**|
> > >
> > >
> > > As highlighted in the table above:
> > > - ***SimPer still substantially outperforms the baselines across smaller window sizes***, even down to a 5 second window. Furthermore, the performance improvements offered by SimPer persist across *different window lengths* and *different datasets*.
> > > - ***Using shorter window lengths for evaluation does not affect the results greatly***. Comparing the performance across a whole video with those using small time windows, while results are generally better with longer time windows, the results do not vary greatly. These experiments indicate that ***the evaluation protocol is robust to time window*** used.
> > >
> > > To summarize, these results show that the FFT evaluation protocol is ***robust***, ***consistent***, and ***reliable*** for evaluating periodic representations, regardless of whether we use shorter window lengths or not. We also highlight that this is expected, as the nearest neighbor & linear probing evaluation protocols are intrinsically different from FFT, but still exhibit the same trend across all datasets.

---

> > > ### Author Response · Authors · 2022-11-30
> > > **Feedback on our latest responses from Reviewer DG5c?**
> > >
> > > We would like to thank the reviewer again for engaging with us and for the questions. We provided our latest responses a week ago (see [detailed responses below](https://openreview.net/forum?id=EKpMeEV0hOo&noteId=-4H8xxJP_CB)), where additional experiments rigorously support our claims and evaluation protocols. We believe the results and clarifications should have successfully addressed all your concerns.
> > >
> > > If the concerns of the reviewer are clarified and the reviewer is convinced of the novelty and completeness of our work, then can we respectfully request the reviewer to update your summary review accordingly?
> > >
> > > Thanks, Authors

---

> > > ### Author Response · Authors · 2022-12-09
> > > **Discussion period ending soon; We would like to hear back from Reviewer DG5c**
> > >
> > > We would like to thank the reviewer again for your time and effort. We have provided [additional results](https://openreview.net/forum?id=EKpMeEV0hOo&noteId=-4H8xxJP_CB) to respond to your latest comments. We believe this evidence should have successfully addressed all your concerns.
> > >
> > > Given there is only **3 days left** in the discussion period, we wanted to double check if the reviewer had seen our latest comment and whether there were any final clarifications the reviewer would like. If the concerns of the reviewer are clarified and the reviewer is convinced of the novelty and completeness of our work, we'd be grateful if the reviewer could update your review and score to reflect that, so we know that our response has been seen. Once again, many thanks for your time and dedication to the review process, we are extremely grateful.

---

> > > > ### Comment · Reviewer_DG5c · 2022-12-09
> > > > **Thanks!**
> > > >
> > > > It is indeed interesting to note that the method performs well on small window evaluation protocol. Given the time constraints, I don't think authors would be able to do these experiments on larger datasets such as the V4V dataset or the VIPL-HR dataset.
> > > >
> > > > However, I thoroughly appreciate these efforts from the authors. I give authors the benefit of doubt and I will increase the score to an 8.

---

> ### Author Response · Authors · 2022-11-17
> **Response to Reviewer DG5c (3/4)**
>
> > *It is not clear if the evaluation protocol is good enough for representations. Especially, for computing a single HR value over a large window. In other words, even if the representations are very bad for large portion of the video, the evaluation protocol would not be able to detect it.*
>
> We respectfully disagree with this point. For feature evaluation, we actually employ **three different standard protocols**: (1) ***FFT***, (2) ***linear probing***, and (3) ***nearest neighbor***. The FFT and linear probing approaches directly evaluate whether the dominating periodicity (frequency) is the target frequency, while the nearest neighbor approach uses the high-dimensional distance between feature vectors to assign the prediction with the label of the most similar feature in the training data.
>
> Regarding your specific comments, in the context of physiological sensing, we agree that using the FFT for feature evaluation is “computing a single HR value over a large window”. However, this is a ***very commonly*** adopted form of evaluation. Almost ***all*** papers reporting results on UBFC and PURE do so based on video-level HR estimation [4, 8, 9, 10]. We argue that if the representations are very bad for a portion of the time window, FFT would not be able to capture the underlying frequency precisely [4, 8] (i.e., FFT is not entirely robust to noise in the PPG waveform). **Moreover**, we also test using the nearest neighbor protocol, which directly compares **each of the components** in the representation; this by definition does not ignore portions of the signal that are corrupted and would be affected by feature representations that are corrupted by noise. **Last but not least**, the FFT results are well correlated with the nearest neighbor (1-NN) results across **all** datasets and methods (Table 2-7), confirming that both of the protocols are able to detect good/bad features. We argue that this provides empirical evidence that the potential issue raised by the reviewer does not exist.
>
> We would like to emphasize that our protocol is designed for **generic** periodic feature evaluation and not for specific tasks or domains. As a result, we tried to strictly follow other examples in the literature to evaluate the learned representations. We’d like to briefly highlight the qualitative and quantitative evidence that this evaluation provided:
>
> **Quantitatively**:
> 1. We followed the SSL literature [5-7] to evaluate the fixed representations through widely accepted and standard protocols: ***FFT***, ***linear probing***, and ***nearest neighbor***
> 2. We followed the SSL literature [5-7] to also use full network fine-tuning for final performance evaluation.
>
> **Qualitatively**:
> 1. We used UMAP to directly visualize the learned feature space by projecting into a 2D Cartesian space (**Figure 1, 4, 5**)
> 2. We further visualized the learned 1D feature to directly assess the feature quality compared with ground truth (**Figure 8, Appendix D.7**).
>
> To summarize, we believe the evaluation protocols are good enough for periodic representations. We certainly welcome any further *specific* and *constructive* suggestions from the reviewer on the evaluation protocol.
>
> &nbsp;
>
>
> > *Additional comparisons with baselines: The results on known traditional methods are better on SCAMPS dataset than the proposed method. These baseline methods have not been to be included in this paper. It would be beneficial to the reader to have a comparison in the paper*
>
> Thanks for your suggestion. As [Reviewer pX9g](https://openreview.net/forum?id=EKpMeEV0hOo&noteId=_0md3jBaKf) pointed out, SimPer is likely **orthogonal** to many advanced techniques, including better architectures, pre-processing methods, etc. We appreciate that in the physiological sensing domain, the main advances in the field do often stem from better backbone architectures and network components [4, 9, 10]. We have added results on SCAMPS below using recent SOTA backbones [9, 10] as well as our SSL technique:
>
> | | MAE ($\downarrow$)| MAPE ($\downarrow$)|
> | - | :-: | :-: |
> |Supervised (Conv3D) |3.61| 5.33|
> |SSL in [8] (Conv3D) |3.51 |5.15 |
> |SimPer (Conv3D) |**2.94** |**4.35**|
> |TS-CAN [9] |2.86 |4.32|
> |SimCLR (TS-CAN) |2.97 |4.47|
> |SimPer (TS-CAN) |**2.44** |**4.09**|
> |EfficientPhys [10] |2.42 |4.10|
> |SimCLR (EfficientPhys) |2.56 |4.17|
> |SimPer (EfficientPhys) |**1.96** |**3.45**|
>
> These results clearly demonstrate that SimPer can improve upon SOTA methods. With recent architectures specialized for learning physiology from videos, SimPer consistently boosts the performance when jointly applied with SOTA techniques [9, 10].
>
> Together with the suggestions from [Reviewer pX9g](https://openreview.net/forum?id=EKpMeEV0hOo&noteId=_0md3jBaKf), we have added additional results and further discussions to **Appendix D.5 and Table 20** to address your concern about the use of other baselines on the SCAMPS dataset, and the compatibility of SimPer with them.

---

> ### Author Response · Authors · 2022-11-17
> **Response to Reviewer DG5c (2/4)**
>
> > *There has already been self-supervised learning approaches in the area of human physiology. This submission does not compare with this ICCV paper.*
>
> Thank you for the suggestion. We agree that providing more baselines will further strengthen the paper, and we provide further discussions and results as follows.
>
> First, we would like to emphasize that the focus of this paper is to provide a **generalized SSL framework of periodic targets**. The mentioned paper [8] studies a specific SSL method tailored for video-based human physiological measurement, and as a result, many of the proposed techniques in [8] only apply to that specific domain (e.g., the face detector, the saliency sampler, and the strong assumptions that are derived from the application context - cf. Table 1 in [8]). However, we do agree that it is possible to extend the SSL loss proposed in [8] (which is a variant of a triplet loss) to other general periodic learning domains.
>
> To confirm the superiority of SimPer over [8], we demonstrate that
> 1. directly compared to [8] on *human physiology datasets*, SimPer achieves significantly and consistently better results under the same setups
> 2. applying the SSL loss in [8] to *other periodic learning domains* further demonstrates the superior performance of SimPer over [8].
>
> ***First***, we directly compare SimPer to [8] on human physiology datasets. All methods use a simple Conv3D backbone on SCAMPS, and a TS-CAN backbone [9] on UBFC/PURE. As the table below indicates, SimPer outperforms [8] across all the tested *human physiology* datasets (i.e., SCAMPS, UBFC, PURE) by a notable margin. We break these results out to show that they hold regardless of whether we include the face saliency module or not.
>
> | |SCAMPS | |UBFC  | |PURE | |
> | - | :-: | :-: | :-: | :-: | :-: | :-: |
> | | MAE ($\downarrow$)| MAPE ($\downarrow$)| MAE ($\downarrow$) |MAPE ($\downarrow$)| MAE ($\downarrow$)| MAPE ($\downarrow$) |
> |*without face saliency module* | | |  | | | |
> |SSL in [8]| 3.53 |5.26| 4.98| 4.61| 4.18| 4.70|
> |SimPer|**3.27**|**4.89**|**4.24**|**3.97**|**3.89**|**4.01**|
> |*with face saliency module* | | |  | | | |
> |SSL in [8]|3.51|5.15|4.88|4.29|4.03|4.28|
> |SimPer|**2.94**|**4.35**|**4.01**|**3.68**|**3.47**|**3.76**|
>
>
> ***Furthermore***, we directly apply the SSL objective in [8] to other domains / datasets involving periodic learning, and show the corresponding results in the table below:
>
> | |RotatingDigits | |Countix  | |LST | |
> | - | :-: | :-: | :-: | :-: | :-: | :-: |
> | | MAE ($\downarrow$)| MAPE ($\downarrow$)| MAE ($\downarrow$) |GM ($\downarrow$)| MAE ($\downarrow$)| $\rho$ ($\uparrow$) |
> |Supervised |0.72 |28.96 |1.50 |0.73 |1.54 |0.96|
> |SSL in [8]| 0.70 |28.03 |1.58| 0.81| 1.62 |0.92|
> |SimPer|**0.20**|**14.33**|**1.33**|**0.59**|**1.47**|**0.96**|
>
> The table clearly shows that the SSL objective in [8] does not provide a benefit in other periodic learning domains, and sometimes performs even worse than the vanilla supervised baseline without any SSL. The above results further emphasize the significance of SimPer, which consistently and substantially exhibits strengths in **general periodic learning** across **all domains**.
>
> We have added these additional results and further discussions to **Appendix D.6, Table 21, and Table 22** to address your concern about other SSL algorithms tailored for human physiology data. We would like to refer you to the newly added section, where we summarize the above experiments to support our statements.
>
> &nbsp;
>
> > *This paper already explores the idea of maximal cross correlation. (MCC)*
>
> We believe there is a misunderstanding here by the reviewer. We are not claiming to propose / explore maximum cross correlation - in fact, maximum cross correlation has been extensively studied in the past decades [3] and widely leveraged in the context of remote physiological measurement [4].
>
> Rather, the key message here is to propose **a family of feature similarity measures** that is suitable for learning periodic targets in a self-supervised manner. That being said, maximum cross correlation (MXCorr), as discussed in the paper (cf. Section 3.2 “Concrete Instantiations”), is **one of the choices for implementation** of the proposed periodic feature similarity. Indeed, we also provide another intuitive choices, i.e., the *Normalized power spectrum density (nPSD)* with both cosine or $L_2$ distance.
>
> In Table 17 in Appendix D.4.3, we show that all of these instantiations can provide good results, where nPSD (with cosine distance) is empirically even better than MXCorr. The results reinforce the argument that it is the notion of **periodic feature similarity** that matters, rather than a specific implementation of the similarity metric.

---

> ### Author Response · Authors · 2022-11-17
> **Response to Reviewer DG5c (1/4)**
>
> Thank you for your thoughtful feedback, your comments have helped us strengthen the evidence for our approach. However, we believe that there are several **important misunderstandings** which we would like to clarify and address point-to-point:
>
> &nbsp;
>
> > *(In “summary of the paper”) The key idea in this paper is to revise the idea of feature similarity for temporal data.*
>
> We would like to clarify that the key idea of SimPer is **not only** to “revise feature similarity for temporal data” but to address **three distinct aspects** of periodic representation learning:
> 1. *How to define positive and negative in periodic learning? (**Section 3.1**)*: Periodicity-variant & periodicity-invariant augmentations w.r.t. the **same** sample;
> 2. *How to define feature similarity in periodic learning? (**Section 3.2**)*: Periodic feature similarity, a **family** of feature similarity measure;
> 3. *How to enable contrastive learning over continuous targets? (**Section 3.3**)*: The generalized InfoNCE loss.
>
> The ideas above were recognized by the other reviewers. We would like to highlight the summaries from [Reviewer pX9g](https://openreview.net/forum?id=EKpMeEV0hOo&noteId=_0md3jBaKf) and [Reviewer y54W](https://openreview.net/forum?id=EKpMeEV0hOo&noteId=-uBvd5KXlV) to support this. We hope that this clarification of the contributions encourages the reviewer to reassess the novelty and significance of our paper.
>
> &nbsp;
>
> > *(In “summary of the paper”) A "random" feature is defined to be a negative feature.*
>
> We would like to clarify that negative features are not **random features**. In fact, it is **exactly the opposite**: the negative samples are **intentionally** constructed directly from the same instance using periodicity-variant augmentations (main paper page 4, Table 1). This distinguishes SimPer from current (contrastive) SSL schemes, in that the design explicitly models what should be the variance in periodic learning (cf. discussions in Section 3.1, Table 1).
>
> Again, we hope the above clarification helps to highlight the novelty and the significance of our paper.
>
> &nbsp;
>
> > *Although the approach is interesting, the experimental section would require additional results to benchmark thoroughly against CVLR. For example, this submission does not include results on Kinetics-400. (as opposed to a subset)*
>
> We would like to clarify that the Kinetics-400 dataset is an **action classification dataset**, where most of the action classes **do not exhibit periodicity** (i.e., the action only occurs once in a given video) [1, 2]. This is exactly why the Countix dataset [1] was proposed. Countix was created from a **subset of videos from Kinetics** to extract meaningful segments of **repeated actions** and corresponding counts, thus resulting in a reasonable dataset for periodic learning / action counting. As such, we believe and argue that testing on the full Kinetics-400 **does not make sense** in the context of periodic learning.
>
> Furthermore, we believe that we have already benchmarked thoroughly against CVRL. As one of the baseline SSL methods, CVRL is tested:
> 1. On **6 datasets** from **4 different domains** (as noted by [Reviewer pX9g](https://openreview.net/forum?id=EKpMeEV0hOo&noteId=_0md3jBaKf))
> 2. Via **feature evaluation** (FFT, linear probing, and nearest neighbor) and **fine-tuning the whole network**
> 3. With results shown both **qualitatively** (i.e., the UMAP of learned features, cf. Fig. 1; the direct visualization of 1D feature, cf. Fig. 8), and **quantitatively** (i.e., Table 2-8, 12, 13).
>
> To summarize, we believe the empirical analysis is comprehensive (as noted by [Reviewer y54W](https://openreview.net/forum?id=EKpMeEV0hOo&noteId=-uBvd5KXlV)) and covers a large array of tasks (as noted by [Reviewer pX9g](https://openreview.net/forum?id=EKpMeEV0hOo&noteId=_0md3jBaKf)). We certainly welcome any further specific suggestions from the reviewer on how the evaluation might be extended.

---

### Official Review · Reviewer_pX9g · 2022-10-25

**Confidence:** 3
**Correctness:** 4
**Technical Novelty And Significance:** 4
**Empirical Novelty And Significance:** 3
**Recommendation:** 10

**Clarity, Quality, Novelty And Reproducibility:**

Please see above for my evaluation of the clarity, quality and novelty of the paper, which are all strengths of the paper in my opinion.

Regarding reproducibility, the submission didn’t include anonymized source code. However, the authors did mention in the appendix their plan to release the code, and given the overall simplicity of the paper, I believe it won’t be too hard to reproduce its results even without the source code.

Below are some additional questions for the authors.
* Given that fractional (<1.0x) and non-integer (e.g. 1.5x rather than 2x, 3x, etc.) frequency scalings seem to be required for augmentation, how are they handled for non-synthetic datasets where frame rates are fixed? Is motion-compensated (rather than naive) frame interpolation used in this paper? How does the quality of frame interpolation affect the results, especially for datasets with larger motions like Countix?
* What’s the shape of the network’s output given a video of length $T$? $\mathbb{R}^T$ or $\mathbb{R}^{T \times H}$ with some hidden dimension $H$? In Sec 4.1, is FFT simply performed over the temporal dimension as 1D FFT? Why is FFT able to predict the underlying frequency given that the actual frequency numbers never seem to be used for feature learning?
* Are Fig 4(a) and 4(b) both showing results for RotatingDigits (according to Sec 4.2) or SCAMPS (according to Sec D.1)? Based on the MAE numbers, Fig 4(b) is likely showing SCAMPS. But given the visual similarity to Fig 1, is Fig 4(a) showing SCAMPS instead of RotatingDigits? Does UMAP create similar visualizations for both datasets?

**Strength And Weaknesses:**

Strengths
+ (Clarity) The paper is very well organized and clearly written. The figures are also well made and very informative.
+ (Novelty) The paper and its proposed techniques are novel as far as I can tell. The topic is clearly important (for its e.g. medical & health applications) yet doesn’t seem to be as popularly studied as other ML/SSL topics.
+ (Quality) The paper is of good quality in my opinion. The algorithm is well designed for periodic signals to maximally extract useful information (via frequency augmentation) to be better learned (via periodic feature similarity & generalized loss). The technical details are all correct as far as I can tell. The empirical evaluation is also comprehensive, covering 6 different SSL (and related) tasks against popular baselines, ~even though the selection of baselines could be further improved (see weaknesses).~
+ (Significance) The proposed techniques are simple, easy to implement and experimentally highly effective, making the algorithm (SimPer) a strong, potentially impactful baseline for future researchers & practitioners to use and/or improve upon.

Weaknesses
- ~(Evaluation, Minor) The algorithm is currently only benchmarked against 4 SSL baselines and a supervised counterpart, not SOTA algorithms on each of the datasets (e.g. the MAE for Countix appears much weaker than Table 7 of Dwibedi et al.). While SimPer is likely orthogonal to those SOTA algorithms and thus might be applied jointly, it’s still non-ideal to completely omit the SOTA baselines and the discussion on how SimPer could further boost the SOTA numbers.~

**Summary Of The Paper:**

The paper presents SimPer, a simple yet effective SSL algorithm specifically designed for periodic tasks (e.g. heart rate estimation & exercise repetition counting through video data, temperature forecasting through satellite data). With the newly proposed techniques including periodicity-variant (i.e. input frequency) augmentation, periodic feature similarity, and generalized (weighted) contrastive loss, SimPer demonstrates impressive results in a large array of tasks including SSL (Table 2 to 7), fine-tuning (Table 8), low-shot learning (Fig 4), transfer learning (Table 9), zero-shot generalization (Table 10), robustness to spurious correlations (Fig 6), etc. against existing SSL algorithms (SimCLR, MoCo v2, BYOL and CVRL) and supervised training, on 6 datasets (2 synthetic) from 4 different domains.

**Summary Of The Review:**

In my opinion, since the paper is strong in ~nearly~ all aspects (clarity, novelty, quality and significance, ~except for minor weakness in evaluation~) with sufficiently impressive results, I recommend its acceptance to the conference.

---

> ### Author Response · Authors · 2022-11-17
> **Response to Reviewer pX9g (2/2)**
>
> > *Given that fractional (<1.0x) and non-integer (e.g. 1.5x rather than 2x, 3x) frequency scalings seem to be required for augmentation, how are they handled for non-synthetic datasets where frame rates are fixed? Is motion-compensated (rather than naive) frame interpolation used? How does quality of frame interpolation affect results, especially for datasets with larger motions like Countix?*
>
> We would like to clarify that **our speed augmentations can be applied to any video**, not just those in a synthetic dataset. The video frames are interpolated in the time domain with a continuous scale factor. This can be implemented in modern deep learning frameworks without additional computational burden (e.g., `tfp.math.interp_regular_1d_grid` in TensorFlow, `torch.nn.functional.interpolate` in PyTorch).
>
> In our analyses we did not use motion compensation. We observed that the vanilla frame interpolation already gave reasonable enough visual results and good final performance, and hence, was fixed throughout to keep the overall scheme simple and efficient.
>
> Nevertheless, we agree with the reviewer that frame interpolation with motion compensation could potentially improve the results on certain datasets with large motions. We will add this good suggestion to the revised paper to help others building on this work.
>
> &nbsp;
>
> > *What’s the shape of the network’s output given a video of length T? $\mathbb{R}^{T}$ or with some hidden dimension?*
>
> The output feature shape is fixed to be $\mathbb{R}^{T}$ for all experiments and SSL methods throughout the paper. Such a representation is standard for the periodic tasks used in the paper [1-4]. This setting is also compatible with the existing (contrastive) SSL algorithms and follows the original setups in the literature [5-7]. We also note that SimPer can be easily extended to high-dimensional features (in addition to the time dimension) by simply averaging across other dimensions (cf. Discussions in main paper, page 4, “Concrete Instantiations”).
>
> &nbsp;
>
> > *In Sec 4.1, is FFT simply performed over the temporal dimension as 1D FFT? Why is FFT able to predict the underlying frequency given that the actual frequency numbers never seem to be used for feature learning?*
>
> Yes, your understanding is correct; since we operate over 1D features, the FFT is performed over the temporal dimension. We extract the frequency simply using an argmax over the power spectrum of the FFT on the embedded features. Thus, no additional parameters are learned for computing the frequency estimate.
>
> To obtain the frequency from the FFT, we only need the frame rate (sampling frequency) of the video, which is available with the input data. As a special case, for the Countix dataset, since the count inside a video is independent of its sampling frequency, we do not even need the frame rate of the videos to obtain the actual count of events.
>
> &nbsp;
>
> > *Are Fig 4a and 4b both showing results for RotatingDigits (according to Sec 4.2) or SCAMPS (according to Sec D.1)? Is Fig 4a showing SCAMPS instead of RotatingDigits? Does UMAP create similar visualizations for both datasets?*
>
> Thank you for pointing this out, and we apologize if this was not clear. Your understanding is correct: Fig 4(a) is for RotatingDigits, which keeps the visualization comparable and coherent throughout the paper; Fig 4(b) however is for SCAMPS, where the caption also indicates that Table 12 & Appendix D.1 provides the full results.
>
> The rationale of using quantitative numbers on SCAMPS is just to show results on a more realistic task. The trend remains the same if we compute quantitative results on RotatingDigits (MAE, $\downarrow$):
>
> | |100%|50%|20%|10%|5%|
> | - | - | - | - |:-:|:-:|
> |Supervised|0.72 |0.86 |1.13|1.87|2.69|
> |SimCLR|0.69|0.77|1.08|1.60|2.34|
> |SimPer|**0.20** |**0.31** |**0.42** |**0.66** |**0.89**|
>
> The UMAP visualization on SCAMPS is similar to that on RotatingDigits, while qualitatively the visual pattern might not seem quite as clear this is due to SCAMPS having greater variance over all. We provide the visualization results in [[this anonymous link]](https://drive.google.com/file/d/1vc5JyosSg2V9D52Q6cYWruklVn_aNwJa/view?usp=sharing) to support the claim and have revised the caption of Figure 4 to clarify this point.
>
> &nbsp;
>
> ---
> References:
> 1. Dwibedi et al. Counting out time: Class agnostic video repetition counting in the wild. CVPR 2020.
> 2. McDuff. Camera measurement of physiological vital signs. ACM Computing Surveys, 2021.
> 3. Liu et al. Multi-task temporal shift attention networks for contactless vitals measurement. NeurIPS 2020.
> 4. Liu et al. EfficientPhys: Enabling simple, fast and accurate camera-based vitals measurement. WACV 2023.
> 5. He et al. Momentum contrast for unsupervised visual representation learning. CVPR 2020.
> 6. Chen et al. A simple framework for contrastive learning of visual representations. ICML 2020.
> 7. Qian et al. Spatiotemporal contrastive video representation learning. CVPR 2021.

---

> > ### Comment · Reviewer_pX9g · 2022-11-22
> > **Re: Response**
> >
> > I appreciate the additional results & detailed feedback provided by the authors, which have more than sufficiently addressed my (and in my opinion, other reviewers’) concerns. I’ve increased my score to 10, as the revised manuscript does stand out among all the ICLR/NeurIPS/ICML papers I’ve reviewed and accepted in the past and should deserve to be highlighted by the conference given its simplicity, versatility and effectiveness.

---

> > > ### Author Response · Authors · 2022-11-23
> > > **Thank you for your encouraging remarks!**
> > >
> > > Thank you very much for your encouraging remarks! We are delighted to learn that our responses have adequately addressed your concerns. Please feel free to let us know if there are other clarifications or experiments we can offer. We would like to thank you again for your time and constructive feedback.

---

> ### Author Response · Authors · 2022-11-17
> **Response to Reviewer pX9g (1/2)**
>
> Thank you very much for acknowledging the novelty and the contributions of our work! We are glad that you found the paper interesting, well-organized and clearly written. In the following, we address your concerns in detail. We hope that these will further clarify our work and lead to a favorable increase of the score.
>
> &nbsp;
>
> > *(Evaluation, Minor) The algorithm is currently only benchmarked against ...... discussion on how SimPer could further boost the SOTA numbers.*
>
> Thank you for your insightful comment. We agree that this will further strengthen the paper, and we provide further discussions and results as follows.
>
> First, before delving into the numbers, we would like to clarify that the experiments in [1] (i.e., Table 7) are not **directly comparable** to those in our paper (i.e., Table 8), as the setting in [1] is intrinsically different from ours in the following ways:
> 1. **Dataset size**. We used a subset (cf. Appendix B, page 15) of Countix, which filters out videos that have a frame length shorter than 200 (the original uncurated data from YouTube has many videos that are too short). This ensures all SSL methods have a fixed number of frames and features, and the dataset is compatible with any network architecture (i.e., ResNet-3D). As such, the training dataset size used in our paper (cf. Table 11) is smaller (i.e., ~40%) than the original Countix dataset (cf. Table 1 in [1]), this was an unavoidable feature of the design.
> 2. **Network architecture**. In [1], the authors proposed a new architecture (i.e., RepNet) that is composed of a ResNet-50 encoder and a Transformer based predictor, which has **3x the number of parameters** compared to the architecture used in our experiments (ResNet-18).
> 3. **ImageNet pre-training**. In [1], the RepNet model’s backbone (i.e., the ResNet-50 encoder) is pre-trained on ImageNet. This effectively enables the network to benefit from more samples and starting from better representations.
> 4. **Stratified training strategies + (camera motion) augmentations**. Last but not least, compared to our fine-tuning stage with a standard training setup, [1] used a more complex training strategy and customized augmentations for this specific task.
>
> Now that we have clarified the differences, let us present results using RepNet [1].  To ensure a fair comparison, we use the open-source code from [1] to train their model on the Countix dataset used in our paper, with all settings used in the original paper, but without ImageNet pre-training (thus keep the number of training samples the same for all methods). The results are as follows:
>
> | |MAE ($\downarrow$)|GM ($\downarrow$)|
> | - | :-: | :-: |
> |RepNet [1]|1.03|0.41|
> |SimCLR (w/ RepNet)| 1.06|0.43|
> |SimPer (w/ RepNet)|**0.72**|**0.22**|
>
> As the table highlights, SimPer does outperform RepNet. Furthermore, we also computed results of RepNet using an ImageNet pre-trained encoder (which is effectively trained on a larger set of data), the MAE of RepNet was still **0.79** (GM 0.27), which is worse than the results for `RepNet + SimPer` even without using that external data. These results show that SimPer has robust performance and importantly **complements** the performance of SOTA models. When used together, SimPer and RepNet lead to the best results.
>
> **In addition**, in the physiological sensing domain (i.e., for datasets: SCAMPS, UBFC, and PURE), the main advances in the field have stemmed from better backbone architectures and network components [2-4]. For SCAMPS, since it is a synthetic dataset, we employed a simple 3D ConvNet (cf. Main paper page 6); as for real datasets UBFC and PURE, we used a more advanced backbone model [3]. To further demonstrate that SimPer can improve upon SOTA methods, we employ a recent architecture specialized for learning physiology from videos [4]. The results on the above three datasets are shown below, where SimPer consistently boosts the performance when jointly applied with the SOTA architecture [4].
>
> | |SCAMPS| |UBFC| |PURE| |
> | - | :-: | :-: | :-: | :-: | :-: | :-: |
> | |MAE|MAPE|MAE|MAPE|MAE|MAPE|
> |EfficientPhys [4]| 2.42|4.10|4.14|3.79|2.87|2.89|
> |SimCLR (w/ EfficientPhys)|2.56|4.17 |4.31 |4.02| 2.94|3.25|
> |SimPer (w/ EfficientPhys) |**1.96**|**3.45**|**3.27**|**3.06**|**2.29**|**2.21**|
>
> To the best of our knowledge there are no other work using the LST and RotatingDigits datasets, it is not obvious which method would be considered state-of-the-art here; instead, we followed what we regarded as the most reasonable backbone (Appendix B & C.2). We certainly welcome any suggestions, and are happy to benchmark more advanced methods with SimPer.
>
> We have added these additional results and further discussions in **Appendix D.5 and Table 20** to address your concern about the use of potential SOTA algorithms on each dataset, and the compatibility of SimPer with them. We would like to refer you to the newly added section, where we summarize the above experiments to support our statements.

---

### Public Comment · ~Sudhakar_Kumawat3 · 2022-11-15
**Comparison with related works of periodic self-supervised learning?**

It seems that contrastive learning [A,B,C] has already been widely used in periodic self-supervised learning.

[A] The Way to my Heart is through Contrastive Learning: Remote Photoplethysmography from Unlabelled Video, ICCV 2021

[B] Self-supervised Representation Learning Framework for Remote Physiological Measurement using Spatiotemporal Augmentation Loss, AAAI 2022

[C] Contrast-Phys: Unsupervised Video-based Remote Physiological Measurement via Spatiotemporal Contrast, ECCV 2022

It should mention in the related work and highlight the difference and novelty explicitly.

It seems the results in Table 4 and Table 5 are worse than [A][C].

---

> ### Author Response · Authors · 2022-11-17
> **These baselines were not designed as general periodic SSL, but rather tailored for one specific application. Experiments and clarifications provided. [1/2]**
>
> Hi Sudhakar,
>
> Thanks for your interest and comments. Your question is similar to one of the points made by [Reviewer DG5c](https://openreview.net/forum?id=EKpMeEV0hOo&noteId=mxu_k5y3d2)’s on the comparison of our approach to SSL methods in the human physiological measurement domain. Similar to [Reviewer DG5c](https://openreview.net/forum?id=EKpMeEV0hOo&noteId=mxu_k5y3d2), we would like to provide the following clarifications and additional experiments to address your comments:
>
> &nbsp;
>
> ***1. Does SimPer perform worse than these methods?***
>
> SimPer actually performs **better** than these methods. We would like to clarify that the experiments and results in [A-C] are not **directly comparable** to those in our paper, as the settings are intrinsically different from ours in the following facts: (1) **different backbones**, where we used a smaller backbone with fewer number of parameters [D]; (2) **different preprocessing**, where the video resolution and frame lengths do not match across papers; (3) **whether incorporating domain knowledge**, where [A-C] typically introduces tricks that apply only to the video-based rPPG domain (e.g., the face detector, the saliency sampler, and the strong assumptions that are derived from the application context - cf. Table 1 in [A]).
>
> To provide a fair comparison, we directly compare SimPer to [A, B] on human physiology datasets. All methods use a simple Conv3D backbone on SCAMPS, and a TS-CAN backbone [D] on UBFC/PURE. As the table below indicates, SimPer outperforms [A, B] across all the tested *human physiology* datasets (i.e., SCAMPS, UBFC, PURE) by a notable margin. We break these results out to show that they hold regardless of whether we include the face saliency [A, B] module or not.
>
> | |SCAMPS | |UBFC  | |PURE | |
> | - | :-: | :-: | :-: | :-: | :-: | :-: |
> | | MAE ($\downarrow$)| MAPE ($\downarrow$)| MAE ($\downarrow$) |MAPE ($\downarrow$)| MAE ($\downarrow$)| MAPE ($\downarrow$) |
> |*without face saliency module* | | |  | | | |
> |SSL in [A]| 3.53 |5.26| 4.98| 4.61| 4.18| 4.70|
> |SSL in [B]| 3.71 |5.54 |5.07 |4.88| 4.32| 4.95|
> |SimPer|**3.27**|**4.89**|**4.24**|**3.97**|**3.89**|**4.01**|
> |*with face saliency module* | | |  | | | |
> |SSL in [A]|3.51|5.15|4.88|4.29|4.03|4.28|
> |SSL in [B]| 3.61 |5.40 |5.02 |4.86 |4.07 |4.33|
> |SimPer|**2.94**|**4.35**|**4.01**|**3.68**|**3.47**|**3.76**|
>
> The above results have been integrated into the revised paper in **Appendix D.6**, **Table 21**. We note that reference [C] was published at ECCV this year, which happened in October and was ***after*** the ICLR submission deadline (September). According to the policy of [ICLR review guideline](https://iclr.cc/Conferences/2023/ReviewerGuide), the paper is considered ***contemporaneous*** and *“authors are not required to compare their own work to that paper”*.
>
> &nbsp;
>
> ***2. Are these papers aimed for general periodic SSL?***
>
> The papers you mentioned are in fact **not designed for generic periodic representation learning**, but rather **tailored for the task of video-based human physiology sensing**. We would like to emphasize that the focus of this paper is to provide a **generalized SSL framework of periodic targets**. The mentioned papers [A-C] however, each study specific a SSL method tailored for video-based human physiological measurement, and as a result, many of the proposed techniques in [A-C] only apply to that specific domain (e.g., the face detector, the saliency sampler, and the strong assumptions that are derived from the application context - cf. Table 1 in [A]). Interestingly, even these papers’ titles would suggest this point.
>
> To demonstrate this point, we directly apply the SSL objective in [A, B] to other domains / datasets involving periodic learning (e.g., action counting, satellite sensing), and show the corresponding results in the table below:
>
> | |RotatingDigits | |Countix  | |LST | |
> | - | :-: | :-: | :-: | :-: | :-: | :-: |
> | | MAE ($\downarrow$)| MAPE ($\downarrow$)| MAE ($\downarrow$) |GM ($\downarrow$)| MAE ($\downarrow$)| $\rho$ ($\uparrow$) |
> |Supervised |0.72 |28.96 |1.50 |0.73 |1.54 |0.96|
> |SSL in [A]| 0.70 |28.03 |1.58| 0.81| 1.62 |0.92|
> |SSL in [B]| 0.77 |29.44 |1.68 |0.94 |1.64 |0.89|
> |SimPer|**0.20**|**14.33**|**1.33**|**0.59**|**1.47**|**0.96**|
>
> The table clearly shows that the SSL objectives in the referenced papers do not provide a benefit in other periodic learning domains, and sometimes perform even worse than the vanilla supervised baseline. The above results further emphasize the significance of SimPer, which consistently and substantially exhibits strengths in **general periodic learning** across **all domains**. We have integrated the above results into the revised paper in **Appendix D.6**, **Table 22**.

---

> > ### Author Response · Authors · 2022-11-17
> > **These baselines were not designed as general periodic SSL, but rather tailored for one specific application. Experiments and clarifications provided. [2/2]**
> >
> > ***3. What is the difference and novelty of SimPer compared to these papers?***
> >
> > We believe the novelty and significance of SimPer is well demonstrated and presented both throughout the paper, as well as through the above experiments.
> >
> > First, the empirical results show that SimPer substantially outperforms [A-C] not only in the human physiological sensing domain, but also on all the other periodic learning tasks we tried. This is because they are **intrinsically different**. Again, no prior work has explicitly studied the problem of periodic representation learning. The techniques proposed in this paper are far different from [A, B, C], resulting in the fact that [A, B, C] is only applicable to the human physiology sensing task, but SimPer is applicable to a wide range of tasks of periodic learning.
> >
> > Further, we hope the clarifications above and the contributions recognized by [Reviewer pX9g](https://openreview.net/forum?id=EKpMeEV0hOo&noteId=_0md3jBaKf) and [Reviewer y54W](https://openreview.net/forum?id=EKpMeEV0hOo&noteId=-uBvd5KXlV) highlight why we argue SimPer is superior to existing methods. SimPer achieves substantially better results than existing SSL methods because it “leverages an **inductive bias** to construct suitable *augmentations* and *loss functions*” ([Reviewer y54W](https://openreview.net/forum?id=EKpMeEV0hOo&noteId=-uBvd5KXlV)), and “the algorithm is well designed for periodic signals to maximally extract useful information” ([Reviewer pX9g](https://openreview.net/forum?id=EKpMeEV0hOo&noteId=_0md3jBaKf)).
> >
> > Last but not least, we would like to again emphasize the design philosophy of SimPer:
> > 1. *How to define positive and negative in periodic learning? (**Section 3.1**)*: Periodicity-variant & periodicity-invariant augmentations w.r.t. the **same** sample;
> > 2. *How to define feature similarity in periodic learning? (**Section 3.2**)*: Periodic feature similarity, a **family** of feature similarity measure;
> > 3. *How to enable contrastive learning over continuous targets? (**Section 3.3**)*: The generalized InfoNCE loss.
> >
> > All related ablation studies on these design choices are supported by experiments (see **Appendix D.4.1–D.4.4**), demonstrating that the design choices make SimPer better than existing SSL methods for learning periodic targets. (Other methods did not explore the intrinsic properties of periodic data learning.)
> >
> > &nbsp;
> >
> > **To summarize**:
> > 1. The papers you mentioned are **not** for general periodic self-supervised learning, but rather only **specific** to the human physiology domain;
> > 2. We provide empirical results showing that within a fair (apples to apples) comparison, SimPer outperforms all the mentioned methods on **human physiology datasets**;
> > 3. We further provide empirical results showing that all the mentioned methods do not improve the periodic learning on other tasks, and sometimes even underperform the vanilla supervised training. In contrast, SimPer leads to **substantially improved results on all datasets across all tasks**.
> >
> > We believe the above discussions further strengthen and highlight the novelty and significance of our paper. We are happy to discuss more if you have any further questions.
> >
> > &nbsp;
> >
> > ---
> > References:
> >
> > [A] The Way to my Heart is through Contrastive Learning: Remote Photoplethysmography from Unlabelled Video, ICCV 2021
> >
> > [B] Self-supervised Representation Learning Framework for Remote Physiological Measurement using Spatiotemporal Augmentation Loss, AAAI 2022
> >
> > [C] Contrast-Phys: Unsupervised Video-based Remote Physiological Measurement via Spatiotemporal Contrast, ECCV 2022
> >
> > [D] Liu et al. Multi-task temporal shift attention networks for on-device contactless vitals measurement. NeurIPS 2020.

---

### Author Response · Authors · 2022-11-17
**General Response to All Reviewers**

We thank all the reviewers for acknowledging the contributions of our work and providing insightful comments and suggestions! We are glad to see that the reviewers found that:

1. Our ideas are novel (Reviewer [DG5c](https://openreview.net/forum?id=EKpMeEV0hOo&noteId=mxu_k5y3d2)), simple and easy to implement (Reviewer [pX9g](https://openreview.net/forum?id=EKpMeEV0hOo&noteId=_0md3jBaKf)), well-motivated and explained very clearly (Reviewer [y54W](https://openreview.net/forum?id=EKpMeEV0hOo&noteId=-uBvd5KXlV)).
2. The problem we propose to solve is timely, challenging, and important (Reviewer [pX9g](https://openreview.net/forum?id=EKpMeEV0hOo&noteId=_0md3jBaKf)).
3. The proposed SimPer algorithm is a strong, potentially impactful baseline for future researchers, and is a valuable contribution to the community (Reviewer [pX9g](https://openreview.net/forum?id=EKpMeEV0hOo&noteId=_0md3jBaKf), [y54W](https://openreview.net/forum?id=EKpMeEV0hOo&noteId=-uBvd5KXlV)).
4. The paper shows very impressive, comprehensive and convincing results (Reviewer [pX9g](https://openreview.net/forum?id=EKpMeEV0hOo&noteId=_0md3jBaKf), [DG5c](https://openreview.net/forum?id=EKpMeEV0hOo&noteId=mxu_k5y3d2), [y54W](https://openreview.net/forum?id=EKpMeEV0hOo&noteId=-uBvd5KXlV)).
5. The experiments and ablation studies are well designed and informative (Reviewer [pX9g](https://openreview.net/forum?id=EKpMeEV0hOo&noteId=_0md3jBaKf), [y54W](https://openreview.net/forum?id=EKpMeEV0hOo&noteId=-uBvd5KXlV)).
6. The manuscript is well organized, clearly written, and easy to follow (Reviewer [pX9g](https://openreview.net/forum?id=EKpMeEV0hOo&noteId=_0md3jBaKf), [DG5c](https://openreview.net/forum?id=EKpMeEV0hOo&noteId=mxu_k5y3d2), [y54W](https://openreview.net/forum?id=EKpMeEV0hOo&noteId=-uBvd5KXlV)).

&nbsp;

To address concerns raised by the reviewers, we have made the following changes in our work and updated the manuscript (major changes are highlighted in *red* in the updated paper):

1. We have incorporated the writing and reference suggestions in the updated paper (**Section 4.6**, **Table 21 & 22**, **Appendix D.6**).
2. We have added more SOTA baselines for each dataset, and the compatibility of SimPer with them in **Appendix D.5** and **Table 20**, as suggested by Reviewer [pX9g](https://openreview.net/forum?id=EKpMeEV0hOo&noteId=_0md3jBaKf), [DG5c](https://openreview.net/forum?id=EKpMeEV0hOo&noteId=mxu_k5y3d2).
3. We have added more SSL baselines for the human physiological sensing datasets, and the comparisons and discussions of why SimPer outperforms them in **Appendix D.6**, **Table 21** and **Table 22**, as suggested by Reviewer [DG5c](https://openreview.net/forum?id=EKpMeEV0hOo&noteId=mxu_k5y3d2).
4. We have added an ablation study of sequence lengths over different tasks and corresponding discussion of its implications in **Appendix D.5** and **Table 19**, as suggested by Reviewer [y54W](https://openreview.net/forum?id=EKpMeEV0hOo&noteId=-uBvd5KXlV).

&nbsp;

We hope our responses have addressed all concerns from the reviewers adequately. We thank all reviewers again for their time and feedback, and please feel free to let us know if there are other clarifications or experiments we can offer. We would really appreciate it if the reviewers could consider raising their scores after evaluating our updates.

---

### Decision · Program_Chairs · 2023-01-20

**Decision:**

Accept: notable-top-5%

**Justification For Why Not Higher Score:**

N/A

**Justification For Why Not Lower Score:**

Given the large interest of the community in SSL techniques and the outstanding performance of the proposed approach, this work might have a large impact.

**Metareview: Summary, Strengths And Weaknesses:**

The paper presents an effective SSL algorithm specifically designed for periodic tasks (e.g. heart rate estimation & exercise repetition counting through video data, temperature forecasting through satellite data). The proposed method (SimPer) includes periodicity-variant augmentation, periodic feature similarity, and a generalized (weighted) contrastive loss. SimPer demonstrates impressive results in a large array of tasks.
All reviewers as well as the AC agree that this is an outstanding contribution as it allows extending SSL techniques to a whole range of applications involving periodic/near-periodic data.


**Note From Pc:**

if the above contains the word "oral" or "spotlight" please see: "oral" presentation means -> notable-top-5% and "spotlight" means -> notable-top-25%. As stated in our emails, we are disassociating presentation type from AC recommendations